# MARS-VFL: A Unified Benchmark for Vertical Federated Learning with Realistic Evaluation

**Wei Shen, Weiqi Liu, Mingde Chen, Wenke Huang, Mang Ye**[*]
National Engineering Research Center for Multimedia Software,
School of Computer Science, Wuhan University
{weishen, yemang}@whu.edu.cn

## Abstract

Vertical Federated Learning (VFL) has emerged as a critical privacy-preserving learning paradigm, enabling collaborative model training by leveraging distributed features across clients. However, due to privacy concerns, there are few publicly available real-world datasets for evaluating VFL methods, which poses significant challenges to related research. To bridge this gap, we propose MARS-VFL, a unified benchmark for realistic VFL evaluation. It integrates data from practical applications involving collaboration across different features, maintaining compatibility with the VFL setting. Based on this, we standardize the evaluation of VFL methods from the mainstream aspects of efficiency, robustness, and security. We conduct comprehensive experiments to assess different VFL approaches, providing references for unified evaluation. Furthermore, we are the first to unify the evaluation of robustness challenges in VFL and introduce a new method for addressing robustness challenges, establishing standard baselines for future research.

## 1 Introduction

Vertical Federated Learning (VFL) [18, 64, 37, 69] is a privacy-preserving learning paradigm that involves training models collaboratively with shared samples but distributed features. Typically, it involves an active client that holds the task labels and multiple passive clients that possess the remaining features of each sample. It finds potential in various real-world applications, such as predicting user credit scores using attributes distributed across different platforms (e.g., banks and shopping centers), or making recommendations between different social media. However, due to data privacy concerns, constructing datasets that share private information across platforms is challenging, resulting in a lack of publicly available datasets for evaluating VFL methods. It poses significant challenges for effective evaluation and presents critical obstacles to the development of VFL research.

Existing works have made extensive efforts to address this issue. The most common approach is to use artificial segmentation, such as splitting images or tabular data into several parts [77]. Other studies [62] propose synthesizing data by considering client correlations and feature importance to better approximate practical data distributions in VFL settings. Some works [60, 61] explore scenarios where multiple links exist between related datasets. They collect several datasets with common identifiers to align several records into a single training sample—for example, predicting a house's price by linking it to prices of nearby houses recorded in another related dataset. Other methods [77] leverage multi-modal data (e.g., NUS-WIDE [10], which includes image and text features) to construct evaluations. FedAds [59] utilizes real-world data from Alibaba to build a two-client dataset and releases anonymized features for research purposes. Despite its recent progress, there remain several challenges in establishing effective benchmarks for VFL algorithms: *(1) Realistic evaluations in VFL settings.* To effectively evaluate VFL methods, it is critical to construct evaluations that align

---

[*]Corresponding author. Codes are available at https://github.com/shentt67/MARS-VFL.

with VFL settings, where different feature blocks of the same sample are distributed across multiple clients. *(2) A unified evaluation benchmark.* To enable fair comparisons between different methods, it is important to define standardized evaluation protocols. *(3) A comprehensive benchmark for different VFL methods.* For a broad analysis, it should cover a wide range of the most recent methods and common challenges in VFL.

Beyond these limitations, we introduce MARS-VFL, a benchmark designed for realistic evaluations tailored to VFL. Our motivation stems from the observation that VFL collaborates with different features of the same sample from multiple clients, which naturally aligns with a wide range of real-world applications involving feature-level collaboration. For example, IoT applications often involve cooperation among different devices (e.g., human-body monitoring devices, robotic sensors). In multimodal applications, different modalities may originate from different different clients—for instance, cross-platform collaborations for user analysis, such as recommendations or emotion analysis, may involve platforms that collect rich visual data (e.g., social media or video-sharing platforms) and others that primarily gather textual or tabular data (e.g., financial platforms). In healthcare, hospitals may hold different types of diagnostic information due to disparities in medical resources and technology. These applications involve collaboration between different feature parts of the same sample, closely aligning with the VFL setting and providing references to evaluate different VFL methods. Building on this insight, we incorporate 12 datasets from five application domains: human activity recognition, robotics, healthcare, emotion analysis, and multimedia analysis. We conduct evaluations with realistic data distributions, where features from different devices, domains, or modalities are distributed across different VFL clients, aligning with real-world VFL deployments and enabling effective evaluation of VFL methods.

Based on the aforementioned construction, we provide unified evaluation protocols covering three fundamental aspects of VFL: *Efficiency*, *Robustness*, and *Security*. (1) *Efficiency:* We provide standardized metrics to compare the efficiency of different methods, including main task performance, communication costs, and convergence speed, summarizing the trade-offs inherent in current methods and moving toward more efficient real-world deployments. (2) *Robustness:* We are the first to unify robustness challenges in VFL, including *missing features*, *corrupted features*, and *misaligned features*. We propose a new method to address corrupted and misaligned features and provide baseline implementations to facilitate future research, fostering the development of more resilient VFL systems. (3) *Security:* We evaluate security vulnerabilities in VFL systems under different types of attacks, including inference attacks and backdoor attacks, and assess the effectiveness of various gradient-based defense strategies, offering insights for the secure deployment of VFL systems. The main contributions of MARS-VFL can be summarized as follows:

- MARS-VFL includes 12 representative datasets across five real-world applications: human activity recognition, robotics, healthcare, emotion analysis, and multimedia analysis. It builds evaluations based on realistic data distributions, where client features originate from different devices, domains, or modalities that are closely aligned with VFL settings, enabling effective VFL evaluations.
- MARS-VFL provides a comprehensive evaluation framework, benchmarking recent VFL methods across three critical aspects: efficiency of different methods, robustness under data perturbations, and security against malicious attacks, providing references for unified evaluations.
- MARS-VFL categorizes robustness challenges in VFL, focusing on missing, corrupted, and misaligned features. We also introduce a new baseline method to address corrupted and misaligned features, expanding the current research landscape for building more stable and robust VFL systems.

## 2 Overview of MARS-VFL

### 2.1 Real-World Applications

Vertical Federated Learning (VFL) collaborates on different features of the same sample from multiple clients. Numerous real-world applications involve feature-level collaboration and naturally align with the VFL setting. Motivated by this insight, we integrate several real-world applications that are closely related to VFL settings, constructing realistic evaluations based on these datasets. Table 1 summarizes the statistics of 12 datasets included in MARS-VFL (illustrated in Figure 1), spanning five application domains: human activity recognition, robotics, healthcare, emotion analysis, and multimedia analysis. All of these applications involve collaboration across different features from the

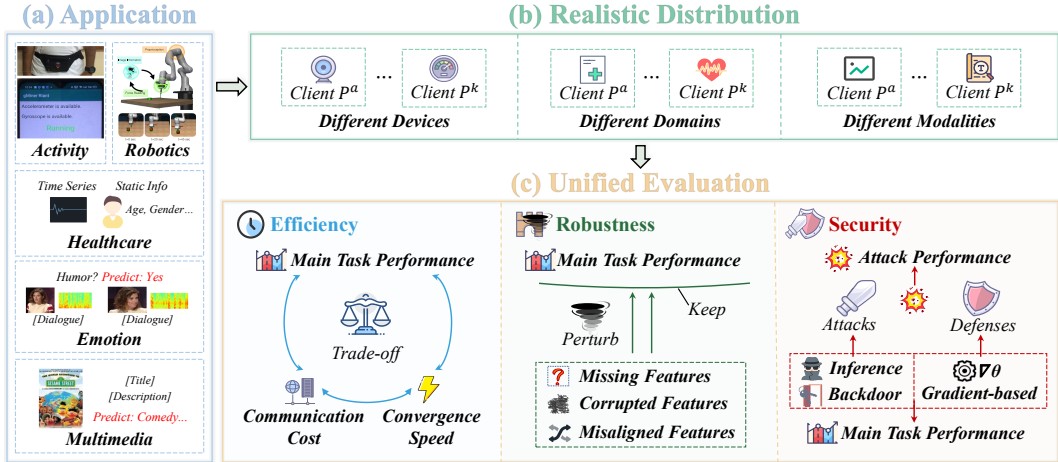

Figure 1: **Framework of MARS-VFL.** (a) MARS-VFL provides a diverse collection of 12 datasets from five real-world applications that align with VFL settings: *human activity*, *robotics*, *healthcare*, *emotion*, and *multimedia*. (b) MARS-VFL follows realistic data distributions across clients in VFL, where data from *different devices*, *data domains*, and *modalities* are distributed among different clients. (c) MARS-VFL offers comprehensive and unified evaluation protocols across three foundational directions: *efficiency*, *robustness*, and *security*, promoting future research in VFL.

Table 1: **Overview of Datasets.** MARS-VFL includes 12 datasets across five different applications.

| Application | Dataset | Number of Clients | Samples | Prediction Task | Target |
|---|---|---|---|---|---|
| HAR | UCI-HAR [2] | 2-client | 10,299 | Activity | 6-class |
| | KU-HAR [50] | 2-client | 20,750 | Activity | 18-class |
| Robotics | MuJoCo [27] | 4-client | 37,990 | Position | 2D Position |
| | VISION&TOUCH [28] | 5-client | 147,000 | Contact State | 2-class |
| Healthcare | MIMIC-III [24] | 2-client | 36,212 | ICD-9 Code | 2-class |
| | PTB-XL [55] | 3-client | 21,700 | ECG Type | 5-class, Multilabel |
| Emotion | UR-FUNNY [20] | 3-client | 16,514 | Humor | 2-class |
| | MUSTARD [7] | 3-client | 690 | Sarcasm | 2-class |
| | CMU-MOSI [70] | 3-client | 2,199 | Emotion | 2-class |
| | CMU-MOSEI [71] | 3-client | 22,777 | Emotion | 2-class |
| Multimedia | NUS-WIDE [10] | 2-client | 116,659 | Content Class | 6-class |
| | MM-IMDB [3] | 2-client | 25,959 | Movie Genre | 23-class, Multilabel |

same sample, which aligns with the VFL setting and provides a basis for evaluating different VFL methods. More details about the datasets can be found in Section B.

**Human Activity Recognition (HAR).** It is a fundamental task in ubiquitous computing and wearable technologies, aiming to predict human actions based on sensor data collected from wearable devices, such as accelerometers and gyroscopes. These sensors capture fine-grained information about body movements and dynamics, enabling a wide range of applications including health monitoring, rehabilitation, and human-computer interaction. MARS-VFL provides realistic evaluations using two public datasets: UCI-HAR [2] and KU-HAR [50]. The accelerometer and gyroscope data are distributed to two separate clients.

**Robotics.** MARS-VFL includes two large-scale robotics datasets: MuJoCo [27] and VISION&TOUCH dataset [28], which capture complex robotic arm operations in real-world environments. These datasets feature robotic systems equipped with diverse sensors—such as cameras and force sensors, each providing different types of data. These sensors can naturally be treated as separate clients in VFL evaluations. In the MuJoCo dataset, the task is to predict the position of the object being manipulated by the robot, while in the VISION&TOUCH dataset, the goal is to predict the robot's contact states. Data from different sensors are distributed to different clients.

**Healthcare.** The application of deep learning in healthcare has shown immense progress across a wide range of fields, enabling breakthroughs in disease diagnosis, patient monitoring, and treatment planning. Medical diagnosis often involves multiple heterogeneous data domains, such as static

patient information (e.g., age and gender), medical imaging data, and time-series physiological signals, each providing complementary information. MARS-VFL integrates two representative datasets: MIMIC-III [24] and PTB-XL [55], where data from different data domains are distributed to different clients to simulate real-world VFL scenarios.

**Emotion Analysis.** Also known as sentiment analysis or affective computing, emotion analysis involves identifying and interpreting emotions from text, speech, or physiological signals. It plays a vital role in various fields, such as mental health monitoring and customer feedback analysis. MARS-VFL includes four datasets—CMU-MOSI [70], CMU-MOSEI [71], UR-FUNNY [20], and MUSTARD [7]—which contain text, video, and audio time-series data for emotion prediction. Data from different modalities are distributed across different clients.

**Multimedia Analysis.** A significant body of research in multimodal learning has been driven by the widespread availability of multimedia data on the internet, such as language, images, video, and audio, leading to substantial progress in content analysis tasks like social media analysis and recommendation systems. MARS-VFL provides evaluations on two datasets: NUS-WIDE [10] and MM-IMDB [3], distributing data from different modalities across different clients.

## 2.2 Unified Evaluations

MARS-VFL offers unified evaluation protocols focusing on three key aspects of VFL: the efficiency of various methods, their robustness under data perturbations, and their security against malicious attacks. It provides a standardized benchmark for comparing different VFL methods.

**Efficiency.** The efficiency of VFL methods stands as one of the primary requirements, particularly in large-scale collaborations where computational resources are often limited. Enhancing the collaboration efficiency of VFL methods is crucial for the practical deployment of VFL systems across various applications. MARS-VFL provides unified evaluations of the efficiency of different methods by standardizing assessments across three key factors: *main task performance (MP)*, *communication costs*, and *convergence speed*, while also exploring the trade-offs between them. For detailed settings, please refer to Section C.1.

**Robustness.** Real-world VFL applications are often impacted by different types of data perturbations, which may occur in different stages, such as data collection, processing, or malicious participants. Investigating the challenges of robustness promotes the development of more stable and robust VFL systems. MARS-VFL firstly unify and provide benchmarks for the robustness of different methods. As shown in Figure 2, there are three key challenges about robustness in VFL: (1) *missing features*, where the clients lack parts of sample features, (2) *corrupted features*, where some samples are affected by data corruptions, and (3) *misaligned features*, where samples are incorrectly aligned. To standardize the evaluation of robustness, we perform evaluations with different perturbation rates of training and test scenarios, examining the *main task performance (MP)* under different perturbations. Please refer to the detailed settings in Section C.2.

**Security.** Despite adhering to the basic privacy protocol, the VFL systems are vulnerable to different security challenges, including the leaking of private data and the destruction of the model behavior, especially with malicious clients whose trustworthiness has not been verified. Investigating security concerns is critical to enhancing reliability and controllability, promoting the safe application of VFL systems. MARS-VFL gives comprehensive evaluations about security issues in VFL, including different attack methods, as well as the defense strategies. There are three main challenges about security in VFL: (1) *label inference attacks*, where usually the passive client, as the attacker, tries to infer the label information of the active client. (2) *feature inference attacks*, where usually the active client acts as the attacker to infer the features of other clients. (3) *backdoor attacks*, where usually the passive client acts as the attacker to induce the target output. MARS-VFL evaluates the *attack performance (AP)* and the *main task performance (MP)* against different gradient-based defense strategies. Please refer to the detailed settings in Section C.3.

## 2.3 A New Baseline for Robustness

MARS-VFL first unifies the robustness challenges across different methods. While several works focus on the challenges of missing features [47, 53], methods against corrupted and misaligned features are lacking. To bridge this gap, we propose a new baseline for robustness in VFL, RVFL, which

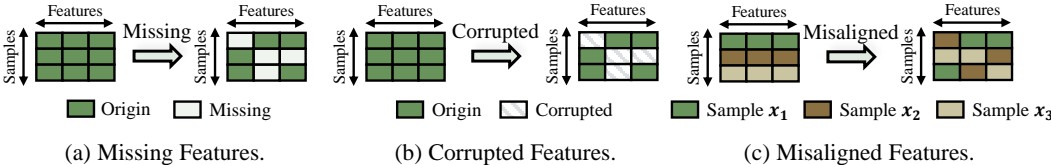

(a) Missing Features.      (b) Corrupted Features.      (c) Misaligned Features.

Figure 2: **Illustration of Robustness.** (a) *Missing Features:* It refers to situations where parts of sample features are missing. (b) *Corrupted Features:* It refers to cases where parts of sample features may be corrupted. (c) *Misaligned Features:* It refers to situations where features across clients are incorrectly aligned with the wrong samples.

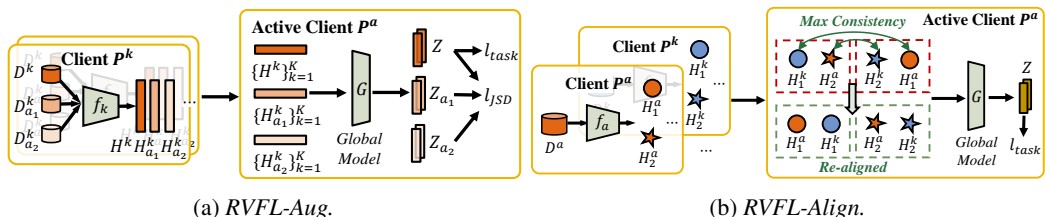

(a) *RVFL-Aug.*          (b) *RVFL-Align.*

Figure 3: **Illustration of RVFL.** (a) *RVFL-Aug* employs different augmentations to learn consistency information and improve robustness against corrupted features. (b) *RVFL-Align* realigns samples by maximizing embedding consistency.

includes two variants, RVFL-Aug and RVFL-Align, designed to handle corrupted and misaligned features, expanding the current research landscape and providing references for future studies.

**Preliminaries.** In VFL, let $K$ denote the number of clients, and $N$ represent the shared samples identified through alignment protocols [18]. Each client $P^k$ maintains a local model $f_k(\cdot; \theta^k)$ to extract embeddings $H_i^k = f_k(x_i^k; \theta^k)$ from the local data $D^k = \{x_i^k\}_{i=1}^K$. Only one client, known as the *the active client* $P^a, a \in \{1, ..., K\}$, holds the task labels $y_i$ with a global model $G(\cdot; \theta^g)$, while the remaining *passive clients* contribute by sending embeddings of shared samples to the active client for model training. Gradients are then distributed back to each client for local updates. The parameters of the overall VFL model are defined as $\Theta = \{\theta^1, ..., \theta^K, \theta^g\}$. Define $\mathcal{L}$ as the loss function, where a cross-entropy loss can be employed for classification tasks. The objective of VFL is to collaboratively update the model parameters $\Theta = \{\theta^1, ..., \theta^K, \theta^g\}$ while preserving data privacy. The basic VFL objective can then be formulated as:

$$\mathcal{L}_{task} = \min_{\Theta} \frac{1}{N} \sum_{i=1}^{N} \mathcal{L}(G(H_i^1, ..., H_i^K; \theta^g), y_i). \qquad (1)$$

This objective function minimizes the average loss $\mathcal{L}$ over all $N$ shared samples, where the global model $G(\cdot; \theta^g)$ makes predictions based on the aggregated embeddings $H_i^k = f_k(x_i^k; \theta^k)$ from each client $P^k$. Since only the active client holds the labels $y_i$, the learning process requires secure collaboration between clients while ensuring no raw data is directly exchanged. The gradients computed from loss functions are backpropagated to update both the local models and the global model iteratively. This formulation enables VFL to leverage distributed features for collaborative model training while preserving data privacy across clients.

**RVFL-Aug.** For corrupted features, we propose utilizing data augmentations to improve the generalization to corrupted data. As shown in Figure 3a, two augmentations of $D^k = \{x_i^k\}_{k=1}^K$ for each client $P^k$ are generated, denoted as $D_{a_1}^k$ and $D_{a_2}^k$. We select a set of general data augmentation operations $\mathcal{A}$ that can be applied to various data types, including random mask, random scale-up, and random scale-down. These augmentations are applied with a random magnitude. Then, an augmentation $a$ is randomly selected from $\mathcal{A}$ and stacked to construct the augmentation sequences $Seq$. The processed data can be formulated as $Seq(x_i^k)$. To preserve the information from the original data, we perform a weighted sum of several sequences, where the number of sequences is $S$, and mix it with the original data. The final augmented data $\tilde{x}_i^k$ can be formulated as follows:

$$\tilde{x}_i^k = \mu \cdot x_i^k + (1 - \mu) \cdot \sum_{i=1}^{S} w_i \cdot Seq(x_i^k), \qquad (2)$$

where a set of weights $(w_1, ..., w_S)$ is randomly sampled from the $Dirichlet(\alpha, \alpha)$ distribution. $\mu$ refers to the weight randomly sampled from the $Beta(\alpha, \alpha)$ distribution. The corresponding embeddings of three datasets, $\{H^k\}_{k=1}^K$, $\{H_{a_1}^k\}_{k=1}^K$, $\{H_{a_2}^k\}_{k=1}^K$ are sent to the active client. The final logit outputs of the embeddings after global model $G(\theta^g)$ are denoted as $Z$, $Z_{a_1}$, and $Z_{a_2}$ respectively. An additional Jensen-Shannon (JS) divergence consistency loss [34] is utilized to ensure the consistency of the model with different augmentations, promoting the robustness to data corruptions. The final objective can be formulated as follows:

$$\mathcal{L}_{aug} = \mathcal{L}_{task} + \lambda \cdot \mathcal{L}_{JS}(Z, Z_{a_1}, Z_{a_2}), \tag{3}$$

where $\lambda$ controls the strength of the JS consistency constraint. The JS divergence is a symmetric version based on Kullback-Leibler (KL) divergence, which ensures the consistency of the outputs with the same samples. We provide extended analysis of RVFL-Aug in Section D.1.

**RVFL-Align.** For misaligned features, we propose to realign the samples based on the embedding consistency. We introduce RVFL-Align, which maximizes embedding consistency between samples from different clients, thereby aligning them optimally. Suppose active client $P^a$ holds the features with the correct corresponding ground truths. This is a reasonable assumption, as the active client coordinates the VFL process and has the label information to verify the correct correspondence. The active client can serve as the anchor to realign the samples. For the embeddings of each passive client $P^k, k \in \{1, \ldots, K-1\}$, compute consistency matrices in active client $P^a$:

$$\mathcal{C}_{ij}^k = \frac{H_i^a \cdot H_j^k}{\|H_i^a\| \|H_j^k\|}, \quad \forall i, j \in (1, N). \tag{4}$$

For embeddings from each passive client $P^k$, solve the linear assignment problem:

$$\max_{M^k \in \{0,1\}^{K \times N}} \sum_{i=1}^N \sum_{j=1}^N M_{ij}^k \cdot \mathcal{C}_{ij}^k, \text{ s.t. } \sum_{i,j} M_{ij}^k = N. \tag{5}$$

The process in Equation (5) rematches the embeddings to maximize consistency, which is more likely to correspond to the same samples. In this way, the sample embeddings are realigned for collaborative learning. After obtaining the matching matrices $M^1, ..., M^{K-1}$, we re-index the samples across all clients to align with the indices of the active client. Specifically, for data of each sample $i$ in the active client $x_i^a$, define the realigned indices $j_k(i)$ of passive client $k$ as:

$$j_k(i) = j \text{ where } M_{i,j}^k = 1. \tag{6}$$

The re-aligned dataset for each client $P^k$ is constructed by re-indexing its samples to match the indices of the active client. The corresponding realigned data can be defined as $\{x_{j_k(i)}^k\}_{i=1}^N$. It is conducted during each forward process, and an extensive analysis is provided in Section D.2.

## 3 Experiments

MARS-VFL provides unified and standardized evaluation protocols for comprehensively assessing the efficiency, robustness, and security of different methods. It serves as a reliable and consistent reference for reproducing experiments and fairly comparing VFL methods. Due to the space limits, we present representative results in the main text, while the complete results are provided in Section E.

### 3.1 Efficiency

We evaluate the efficiency of recent methods including FedBCD [36], C-VFL [6], and EFVFL [54]. FedBCD achieves faster convergence by introducing local updates, while C-VFL and EFVFL reduce communication costs through compression techniques. We assess these methods in terms of main task performance (MP), communication costs, and convergence speed, analyzing the trade-offs among these factors. The detailed experiment settings are provided in Section C.1. A subset of the results is presented here, with additional results provided in Section E.1.

**Best Main Task Performance (Best MP).** In Table 2, we run each method five times and report the mean and standard deviation of the performance. We present the maximum test MP achieved by

Table 2: **Results of Best Main Task Performance (MP (%))**. The main results on eight datasets are reported. 'Best MP' refers to the highest test MP achieved, with the corresponding communication costs and training epochs. The definitions of evaluation metrics are detailed in Section C.1.

| Method | UCI-HAR | | | KU-HAR | | | MIMIC-III | | | PTB-XL | | |
|---|---|---|---|---|---|---|---|---|---|---|---|---|
| | Best MP | Costs (MB) | Epochs | Best MP | Costs (MB) | Epochs | Best MP | Costs (GB) | Epochs | Best MP | Costs (GB) | Epochs |
| Base | 95.03±0.69 | 114.52±16.75 | 128±19 | 82.55±0.58 | 535.72±24.86 | 141±7 | 61.35±0.49 | 41.93±0.07 | 299±1 | **57.41±0.52** | 270.97±33.00 | 70±8 |
| FedBCD [36] | 95.03±0.27 | 78.98±27.75 | **88±31** | **86.07±0.56** | 488.61±59.57 | 129±16 | 61.34±0.36 | 38.32±4.84 | 274±35 | 54.92±0.26 | 100.44±47.20 | 26±12 |
| C-VFL [6] | 95.01±0.34 | **25.58±9.79** | 95±36 | 84.75±0.60 | **145.90±14.14** | 128±12 | 61.49±0.64 | **11.96±0.47** | 285±11 | 53.41±1.02 | **27.33±8.82** | **23±8** |
| EFVFL [54] | **95.21±0.22** | 26.49±10.69 | 98±40 | 85.00±0.45 | 150.69±12.52 | 132±11 | **61.76±0.49** | 12.13±0.48 | 289±11 | 54.53±1.11 | 35.27±4.87 | 30±4 |

| Method | NUS-WIDE | | | MUSTARD | | | UR-FUNNY | | | MM-IMDB | | |
|---|---|---|---|---|---|---|---|---|---|---|---|---|
| | Best MP | Costs (MB) | Epochs | Best MP | Costs (MB) | Epochs | Best MP | Costs (GB) | Epochs | Best MP | Costs (GB) | Epochs |
| Base | **81.53±0.94** | 230.60±31.38 | 7±1 | **57.83±2.26** | 187.29±27.76 | 78±12 | **63.63±0.39** | 2.71±0.90 | 59±20 | 56.63±0.33 | 3.56±0.38 | 60±6 |
| FedBCD [36] | 81.02±0.71 | 128.11±53.59 | 4±2 | 54.49±2.44 | 152.89±47.75 | 64±20 | 63.61±0.17 | 1.00±0.65 | **22±14** | **56.83±0.36** | 0.78±0.11 | 13±2 |
| C-VFL [6] | 80.63±0.82 | **38.43±10.53** | **4±1** | 54.34±2.26 | **36.55±17.20** | 51±24 | 62.95±1.43 | **0.44±0.04** | 32±3 | 56.31±0.22 | **0.23±0.03** | **13±2** |
| EFVFL [54] | 81.21±0.48 | 49.96±15.37 | 5±2 | 56.52±4.05 | 38.99±14.61 | 54±20 | 63.26±0.64 | 0.44±0.21 | 32±15 | 56.42±0.18 | 0.27±0.02 | 15±1 |

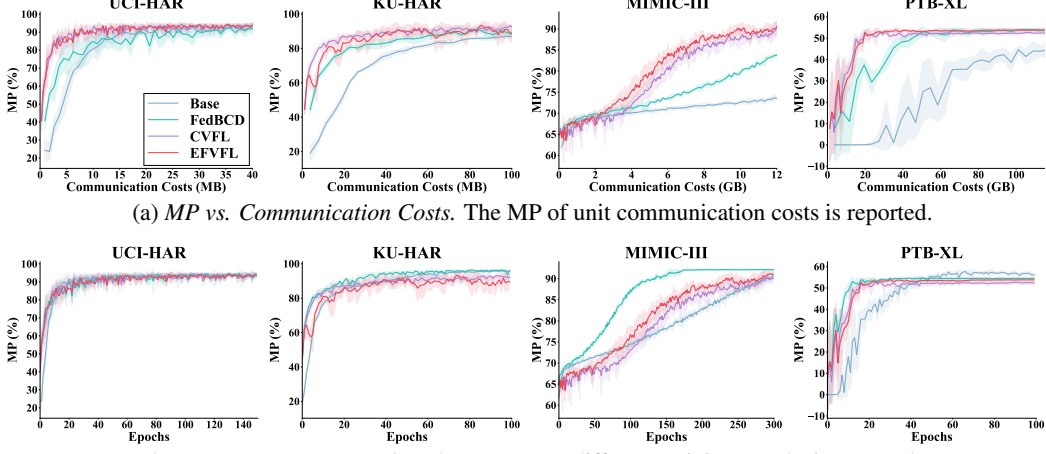

(a) *MP vs. Communication Costs.* The MP of unit communication costs is reported.

(b) *MP vs. Training Epochs.* The MP across different training epochs is reported.

Figure 4: **MP in Training Stages.** The training curves w.r.t communication cost and epochs.

each method, along with the corresponding communication costs and training epochs. Compared to Base (standard VFL without additional operations), FedBCD, C-VFL, and EFVFL achieve lower communication costs and require fewer training epochs to converge, demonstrating improved efficiency. However, in datasets such as NUS-WIDE, MUSTARD, and UR-FUNNY, these efficiency gains come at the expense of MP, raising *a trade-off between MP and communication costs*, as well as *the trade-off between MP and convergence speed (epochs).* Additionally, while C-VFL and EFVFL reduce communication costs through embedding compression, they generally require more training epochs than FedBCD and show decreased performance, due to information loss from compression. This reveals *the trade-off between communication costs and convergence speed.*

**Performance by Communication Costs.** As shown in Figure 4a, we evaluate the efficiency of each method by comparing their MP under the same communication costs, highlighting the relative efficiency of different approaches. For the UCI-HAR and KU-HAR datasets, which are split into training and test sets, we report the test MP over epochs. For the MIMIC-III and PTB-XL datasets, which are split into training, validation, and test sets, we report the validation MP over epochs. Compared to Base method, FedBCD, C-VFL, and EFVFL achieve higher performance under the same communication costs, indicating better efficiency and faster convergence. Furthermore, C-VFL and EFVFL outperform FedBCD in MP per unit of communication, demonstrating lower communication costs for achieving the same performance, owing to the use of embedding compression.

**Performance by Epochs.** As illustrated in Figure 4b, we evaluate the efficiency of different methods by comparing their test/validation MP across training epochs. Compared to Base, FedBCD, C-VFL, and EFVFL reach stable performance in fewer epochs, indicating faster convergence. Moreover, the MP of FedBCD is generally higher than that of C-VFL and EFVFL across epochs, suggesting that embedding compression introduces additional optimization challenges and slower convergence.

Table 3: **Results with Missing Features.** MP is reported with $r_a, r_b = 0, 0.2, 0.5$.

| | UCI-HAR (MP: $\mathcal{A}$ (%)) | | | | | | | | | KU-HAR (MP: $\mathcal{A}$ (%)) | | | | | | | | |
|---|---|---|---|---|---|---|---|---|---|---|---|---|---|---|---|---|---|---|
| Method | $r_a=0$ | | | $r_a=0.2$ | | | $r_a=0.5$ | | | $r_a=0$ | | | $r_a=0.2$ | | | $r_a=0.5$ | | |
| | $r_b=0$ | $r_b=0.2$ | $r_b=0.5$ | $r_b=0$ | $r_b=0.2$ | $r_b=0.5$ | $r_b=0$ | $r_b=0.2$ | $r_b=0.5$ | $r_b=0$ | $r_b=0.2$ | $r_b=0.5$ | $r_b=0$ | $r_b=0.2$ | $r_b=0.5$ | $r_b=0$ | $r_b=0.2$ | $r_b=0.5$ |
| Base | 94.67 | 90.77 | 90.50 | 94.47 | 90.23 | 89.55 | 93.93 | 89.65 | 89.48 | 82.24 | 66.41 | 38.12 | 81.49 | 65.93 | 38.48 | 79.98 | 61.78 | 37.61 |
| LEEF-VFL [47] | 94.60 | 90.80 | 91.92 | 95.05 | 91.25 | 91.04 | 95.76 | 90.70 | 90.22 | 82.22 | **75.54** | 43.81 | 81.59 | 74.00 | 43.18 | 80.41 | 72.39 | 42.12 |
| LASER-VFL [53] | **95.88** | **94.33** | **95.37** | **95.64** | **95.01** | **95.32** | **95.76** | **94.91** | **94.79** | **82.69** | 75.42 | **65.03** | **81.83** | **74.82** | **63.38** | **80.55** | **74.57** | **64.06** |

| | MM-IMDB (MP: $F_1$ (%)) | | | | | | | | | MuJoCo (MP: MSE) | | | | | | | | |
|---|---|---|---|---|---|---|---|---|---|---|---|---|---|---|---|---|---|---|
| Method | $r_a=0$ | | | $r_a=0.2$ | | | $r_a=0.5$ | | | $r_a=0$ | | | $r_a=0.2$ | | | $r_a=0.5$ | | |
| | $r_b=0$ | $r_b=0.2$ | $r_b=0.5$ | $r_b=0$ | $r_b=0.2$ | $r_b=0.5$ | $r_b=0$ | $r_b=0.2$ | $r_b=0.5$ | $r_b=0$ | $r_b=0.2$ | $r_b=0.5$ | $r_b=0$ | $r_b=0.2$ | $r_b=0.5$ | $r_b=0$ | $r_b=0.2$ | $r_b=0.5$ |
| Base | 56.35 | 31.79 | 19.99 | 55.26 | 30.43 | 19.88 | 54.95 | 30.42 | 19.89 | 0.016 | 4.998 | 7.779 | 0.021 | 4.984 | 7.783 | 0.063 | 4.997 | 7.785 |
| LEEF-VFL [47] | 56.53 | 35.77 | 21.22 | 55.29 | 36.22 | 21.82 | 55.04 | 35.33 | 21.35 | 0.013 | 4.322 | 7.741 | 0.016 | 4.713 | 7.743 | 0.015 | 4.731 | 7.754 |
| LASER-VFL [53] | **56.62** | **50.44** | **50.30** | **55.47** | **50.50** | **50.37** | **55.15** | **49.17** | **49.04** | **0.012** | **0.103** | **0.103** | **0.014** | **0.109** | **0.109** | **0.014** | **0.107** | **0.108** |

Table 4: **Results with Corrupted Features.** MP is reported with $r_a, r_b = 0, 0.5, 0.8$.

| | UCI-HAR (MP: $\mathcal{A}$ (%)) | | | | | | | | | KU-HAR (MP: $\mathcal{A}$ (%)) | | | | | | | | |
|---|---|---|---|---|---|---|---|---|---|---|---|---|---|---|---|---|---|---|
| Method | $r_a=0$ | | | $r_a=0.5$ | | | $r_a=0.8$ | | | $r_a=0$ | | | $r_a=0.5$ | | | $r_a=0.8$ | | |
| | $r_b=0$ | $r_b=0.5$ | $r_b=0.8$ | $r_b=0$ | $r_b=0.5$ | $r_b=0.8$ | $r_b=0$ | $r_b=0.5$ | $r_b=0.8$ | $r_b=0$ | $r_b=0.5$ | $r_b=0.8$ | $r_b=0$ | $r_b=0.5$ | $r_b=0.8$ | $r_b=0$ | $r_b=0.5$ | $r_b=0.8$ |
| Base | **94.67** | 88.60 | 86.16 | **94.27** | 87.61 | 84.12 | 93.86 | 87.95 | 84.15 | 82.24 | 69.98 | 63.76 | 79.37 | 71.37 | 66.72 | 76.58 | 69.73 | 66.12 |
| RVFL-Aug | 93.99 | **90.23** | **88.19** | 94.13 | **88.63** | **85.99** | **94.91** | **88.29** | **85.37** | **82.31** | **70.77** | **64.60** | **80.12** | **72.46** | **67.52** | **77.04** | **70.53** | **66.67** |

| | MM-IMDB (MP: $F_1$ (%)) | | | | | | | | | MuJoCo (MP: MSE) | | | | | | | | |
|---|---|---|---|---|---|---|---|---|---|---|---|---|---|---|---|---|---|---|
| Method | $r_a=0$ | | | $r_a=0.5$ | | | $r_a=0.8$ | | | $r_a=0$ | | | $r_a=0.5$ | | | $r_a=0.8$ | | |
| | $r_b=0$ | $r_b=0.5$ | $r_b=0.8$ | $r_b=0$ | $r_b=0.5$ | $r_b=0.8$ | $r_b=0$ | $r_b=0.5$ | $r_b=0.8$ | $r_b=0$ | $r_b=0.5$ | $r_b=0.8$ | $r_b=0$ | $r_b=0.5$ | $r_b=0.8$ | $r_b=0$ | $r_b=0.5$ | $r_b=0.8$ |
| Base | 56.35 | 34.86 | 33.48 | 39.46 | 31.31 | 27.95 | 37.47 | 25.49 | 15.26 | 0.016 | 0.126 | 0.212 | **0.014** | 0.032 | **0.033** | 0.018 | 0.040 | 0.043 |
| RVFL-Aug | 55.68 | **42.59** | **42.21** | **44.22** | **41.26** | **39.63** | **43.27** | **34.46** | **24.72** | **0.013** | **0.110** | **0.196** | 0.015 | **0.028** | 0.040 | **0.017** | **0.031** | **0.041** |

Table 5: **Results with Misaligned Features.** MP is reported with $r_a, r_b = 0, 0.8, 1$.

| | UCI-HAR (MP: $\mathcal{A}$ (%)) | | | | | | | | | KU-HAR (MP: $\mathcal{A}$ (%)) | | | | | | | | |
|---|---|---|---|---|---|---|---|---|---|---|---|---|---|---|---|---|---|---|
| Method | $r_a=0$ | | | $r_a=0.8$ | | | $r_a=1$ | | | $r_a=0$ | | | $r_a=0.8$ | | | $r_a=1$ | | |
| | $r_b=0$ | $r_b=0.8$ | $r_b=1$ | $r_b=0$ | $r_b=0.8$ | $r_b=1$ | $r_b=0$ | $r_b=0.8$ | $r_b=1$ | $r_b=0$ | $r_b=0.8$ | $r_b=1$ | $r_b=0$ | $r_b=0.8$ | $r_b=1$ | $r_b=0$ | $r_b=0.8$ | $r_b=1$ |
| Base | **94.67** | 90.87 | 89.01 | 90.87 | 87.75 | 86.56 | 90.60 | 85.23 | 84.86 | **82.24** | 70.48 | 70.17 | 70.39 | 68.12 | 68.31 | 70.29 | 47.40 | 40.24 |
| RVFL-Align | 94.47 | **90.91** | **91.01** | **91.08** | **91.08** | **91.04** | **90.80** | **91.11** | **91.08** | 81.20 | **70.77** | **70.65** | **71.16** | **70.75** | **70.77** | **70.72** | **68.17** | **62.31** |

| | MM-IMDB (MP: $F_1$ (%)) | | | | | | | | | MuJoCo (MP: MSE) | | | | | | | | |
|---|---|---|---|---|---|---|---|---|---|---|---|---|---|---|---|---|---|---|
| Method | $r_a=0$ | | | $r_a=0.8$ | | | $r_a=1$ | | | $r_a=0$ | | | $r_a=0.8$ | | | $r_a=1$ | | |
| | $r_b=0$ | $r_b=0.8$ | $r_b=1$ | $r_b=0$ | $r_b=0.8$ | $r_b=1$ | $r_b=0$ | $r_b=0.8$ | $r_b=1$ | $r_b=0$ | $r_b=0.8$ | $r_b=1$ | $r_b=0$ | $r_b=0.8$ | $r_b=1$ | $r_b=0$ | $r_b=0.8$ | $r_b=1$ |
| Base | **56.35** | 52.11 | 51.43 | 53.25 | 51.76 | 50.08 | 51.33 | 51.43 | 49.93 | **0.016** | 0.217 | 0.218 | 0.183 | 0.264 | 0.304 | 0.183 | 0.335 | 0.410 |
| RVFL-Align | 55.84 | **52.17** | **51.93** | **54.94** | **51.89** | **51.78** | **52.55** | **51.67** | **51.72** | 0.017 | **0.165** | **0.173** | **0.142** | **0.181** | **0.182** | **0.182** | **0.183** | **0.183** |

## 3.2 Robustness

We evaluate the robustness of several methods, including LEEF-VFL [47], LASER-VFL [53], and the proposed RVFL (RVFL-Aug and RVFL-Align). We report the main task performance under different perturbation rates applied to the training, validation, and test sets. In each experiment, the training and validation sets share the same perturbation rates, while the test set is evaluated under different perturbation rates. Detailed settings and additional results are provided in Section C.2 and Section E.2, respectively.

**Missing Features.** Following the settings in prior work [53], we evaluate the performance of LEEF-VFL [47] and LASER-VFL [53] under different missing rates in the training, validation, and test sets. Denote $r_a$ as the missing rate in the training/validation datasets, and $r_b$ as the missing rate in the test dataset, representing the probability that each feature part is missing. As shown in Table 3, both LEEF-VFL and LASER-VFL outperform Base across different missing rates, with LASER-VFL achieving superior performance due to its adaptation to missing features in the test set.

**Corrupted Features.** We evaluate the performance of RVFL-Aug under different corruption rates, where $r_a$ denotes the corruption rate for the training and validation datasets, and $r_b$ for the test dataset. These rates represent the proportion of corrupted feature parts, with Gaussian noise added to randomly selected features to simulate corruption (detailed in Section C.2). This setup can be easily extended to support various types of corruption. As shown in Table 4, RVFL-Aug consistently outperforms Base under different corruption settings, demonstrating improved robustness through consistency learning across augmentations.

**Misaligned Features.** We evaluate RVFL-Align under different misalignment rates, where $r_a$ and $r_b$ denote the proportions of misaligned samples in the training/validation and test datasets, respectively. As shown in Table 5, RVFL-Align consistently outperforms the Base method across different levels of misalignment, demonstrating its effectiveness in realigning samples through maximum consistency.

## 3.3 Security

We examine the performance of different attack methods, including (1) PMC [15] and AMC [15] for label inference attacks. (2) GRNA [38] and MIA [30] for feature inference attack. (3) TECB [8] and LFBA [46] for backdoor attacks. We evaluate four gradient-based defense strategies and report

Table 6: **Performance of Attacks.** The MP and AP are reported.

| Metric | UCIHAR | | | | | | NUSWIDE | | | | | |
| | Label Inference | | Feature Inference | | Backdoor | | Label Inference | | Feature Inference | | Backdoor | |
| | PMC[15] | AMC[15] | GRNA [38] | MIA[30] | TECB [8] | LFBA [46] | PMC[15] | AMC[15] | GRNA [38] | MIA[30] | TECB [8] | LFBA [46] |
|---|---|---|---|---|---|---|---|---|---|---|---|---|
| MP (%) | **94.67** | 92.91 | **93.28** | 92.50 | 89.11 | **92.16** | **81.42** | 80.46 | **81.64** | 81.33 | 74.99 | **81.07** |
| AP (%) | 60.50 | **64.27** | 38.55 | **74.45** | **100** | 99.39 | 57.43 | **60.23** | 36.25 | **99.75** | **100** | 99.93 |

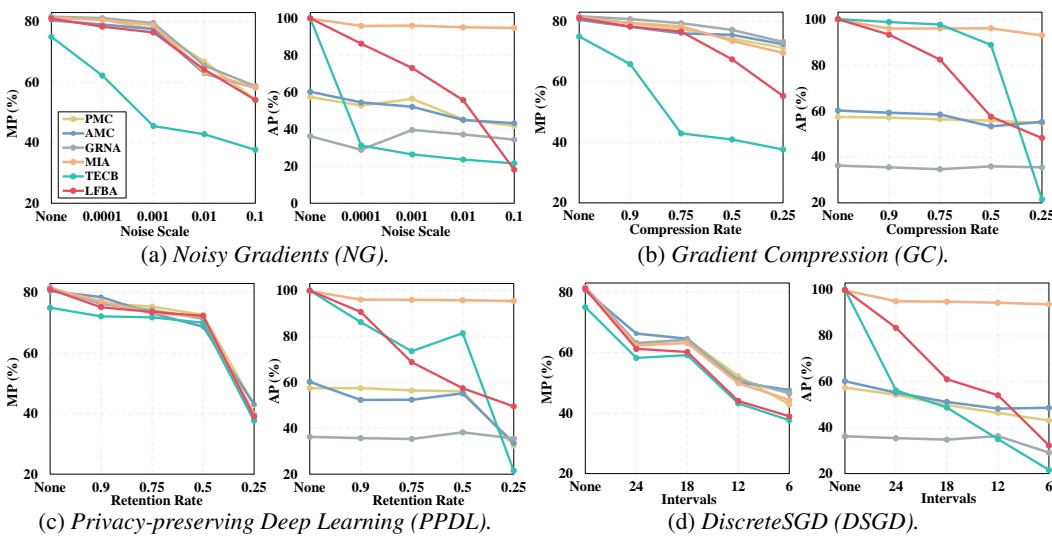

(a) *Noisy Gradients (NG).*

(b) *Gradient Compression (GC).*

(c) *Privacy-preserving Deep Learning (PPDL).*

(d) *DiscreteSGD (DSGD).*

Figure 5: **Performance of Defenses.** The MP and AP are reported on the NUS-WIDE dataset.

the performance of each attack under these defenses. Detailed experimental settings and additional results are provided in Section C.3 and Section E.3, respectively.

**Performance of Attacks.** We evaluate the attack performance (AP) of various attacks, as well as the main task performance (MP) of models under different attacks. The evaluation metrics are detailed in Section C.3. The attack performance on the UCI-HAR and NUS-WIDE datasets is presented in Table 6. As shown in Table 6, different attacks pose significant threats to VFL systems, with label inference attacks achieving over 60% accuracy, feature inference attacks reaching over 90% AP, and backdoor attacks exhibiting strong control with attack success rates exceeding 90%.

**Performance of Defenses.** To investigate potential defenses, we evaluate the AP and MP of attacks under various gradient-based defense strategies, following the setting in [15]. These include Noisy Gradients (NG) [76], Gradient Compression (GC) [35], Privacy-preserving Deep Learning (PPDL) [49], and DiscreteSGD (DSGD) [4, 15]. Details of each method are provided in Section C.3. As shown in Figure 5, we report the AP and MP under different defense parameter settings. These defenses reduce attack performance but at the cost of degrading main task performance, indicating that the evaluated defenses are insufficient to effectively mitigate various attacks, highlighting a potential direction for future research.

## 4 Conclusion

In summary, we introduce MARS-VFL, a comprehensive and systematically designed benchmark that provides realistic and unified evaluation protocols for an in-depth assessment of the key dimensions of efficiency, robustness, and security in VFL systems. To achieve this goal, MARS-VFL integrates a diverse suite of 12 datasets covering five representative real-world applications, thereby enabling evaluations under practical and heterogeneous data distributions. Beyond dataset integration, MARS-VFL makes a notable contribution by establishing the first unified perspective on the broad spectrum of robustness challenges in VFL, and by proposing a new baseline method that effectively mitigates these issues, substantially enriching the current research landscape. Through extensive evaluations across a range of recent and representative methods, MARS-VFL aims to deliver valuable empirical

evidence, guidance, and insights for the community, ultimately supporting the development of more generalizable, robust, and inherently secure VFL systems suitable for real-world deployment.

## Ethic Impacts

**Privacy of Human-derived Data.** This study involves datasets that contain human-derived data, which may raise potential privacy concerns. Specifically, datasets such as MIMIC-III and PTB-XL are derived from real patients and may pose privacy risks even after anonymization. All data used in this work are publicly available, and the participants had provided consent for research use. These datasets were used solely for scientific purposes, and all personally identifiable information was removed by the dataset providers. Ensuring user consent and proper anonymization remains essential for ethical data usage. For real-world deployment of the evaluated VFL methods, privacy-preserving techniques can be readily integrated into the framework to further protect user data [18].

**Attack Methods.** The evaluated attack methods reveal that vertical federated learning systems can be vulnerable to security and privacy risks, such as label inference attacks. These vulnerabilities highlight potential negative societal impacts if exploited, especially in sensitive domains like healthcare, where private information could be inferred from model updates. Recognizing such risks is crucial for developing more robust and privacy-preserving federated learning systems.

**Potential Defenses.** Current defense methods also exhibit limitations. While gradient-based defenses [15] can mitigate certain attacks, they remain ineffective against others, such as embedding reconstruction and membership inference attacks. Advancing defense mechanisms to comprehensively safeguard user data is an important direction for future research.

**Environmental Impact.** The environmental footprint of our benchmarking process is low. Most experiments were performed for research purposes only and are computationally efficient. The runtime and energy consumption remain within a sustainable range for an individual researcher, and we encourage the community to continue exploring energy-efficient federated learning evaluations.

## Acknowledgments and Disclosure of Funding

This work is supported by National Natural Science Foundation of China under Grants (62361166629, 623B2080), the Major Project of Science and Technology Innovation of Hubei Province (2024BCA003, 2025BEA002), and the Innovative Research Group Project of Hubei Province under Grants (2024AFA017). The supercomputing system at the Supercomputing Center of Wuhan University supported the numerical calculations in this paper.

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

# Appendix

## A    Related Work

Federated Learning (FL) was first proposed by Google [39], enabling collaborative model training across different participants without sharing the original data. Based on the distribution of data, FL can be categorized into Horizontal Federated Learning (HFL) [65, 31, 66, 22] and Vertical Federated Learning (VFL) [18, 58, 64, 37, 69]. In HFL, different participants share the same feature space but typically possess their private samples. Each client trains a local model, and the model parameters are then aggregated on the server, thereby avoiding the sharing of raw data while leveraging the resources of all clients. In contrast, in VFL, participants share the same samples but have different parts of the feature space. Only one client possesses the labels for the collaborative tasks, referred to as the active client, while the remaining clients are passive. During training, only the embeddings and gradients are transmitted, ensuring the privacy of the original data. The concept of VFL was introduced by [18] and has since shown promising results and potential in various applications, such as healthcare [21, 63], where different hospitals have different medical data for the same patient, internet recommendation systems [59], and finance [75], where different platforms hold different user data for privacy-preserving collaborations [44, 45, 67, 68], among others [13, 43, 56, 12, 16, 57]. In this work, we propose MARS-VFL, which offers realistic evaluations based on real-world applications that align with VFL settings, as well as a unified benchmark for VFL evaluations. Our benchmark focuses on three key aspects of VFL algorithms: Efficiency, which assesses the training efficiency of different VFL methods; Robustness, which evaluates the stability and generalization of VFL methods under various perturbations; and Security, which addresses privacy concerns arising from malicious attacks and explores potential defenses against these threats. With these directions, MARS-VFL covers a wide range of recent works, providing valuable protocols for evaluating VFL methods and promoting future research.

## B    Datasets and Models

MARS-VFL provides pipelines that span from data processing to model training, as well as unified evaluation protocols and metrics, along with public code to promote future research in VFL. We include different real-world applications and their associated datasets, which align with VFL settings. We provide detailed information about the data distribution of each client, as well as the processing procedures to facilitate the use of the benchmark. Additionally, we provide the specifics of the backbones used for each dataset (the setting is for reproducing the results of the 'Base' method, which represents standard VFL training without additional modifications; the results for other methods are typically the same, unless specified otherwise). For datasets with a validation set, the models with the best validation performance are saved, and the corresponding test performance is reported; for other datasets, the best test performance is reported.

### B.1    Human Activity Recognition

**UCI-HAR** [2]: It is a human activity recognition dataset collected from a group of 30 volunteers, aged 19-48 years. Each participant performed six activities: walking, walking upstairs, walking downstairs, sitting, standing, and laying. Each volunteer wore a smartphone on their waist to record the activities. Using the smartphone's embedded accelerometer and gyroscope, the 3-axial linear acceleration and 3-axial angular velocity were captured at a constant rate of 50 Hz. The sensor signals (accelerometer and gyroscope) were pre-processed with noise filters and then sampled in fixed-width sliding windows of 2.56 seconds with $50\%$ overlap, resulting in a total of $10,299$ samples. The task is to predict the class of human activities, with the classification accuracy reported.

*Dataset Processing:* The data from the accelerometer and gyroscope are distributed across two different clients ($K = 2$). We use the predefined split for the training and test sets, where $70\%$ of the volunteers were randomly selected to generate the training data and $30\%$ were used for the test data.

*Models:* The models and corresponding hyperparameters used for the UCI-HAR dataset are shown in Table 7. For the local models on each client, we use a three-layer MLP to extract the original features into 16-dimensional embeddings. A two-layer MLP is then used in the active client to make

Table 7: **Models and Hyperparemeters for UCI-HAR Dataset.**

| Module Description | | Hyperparameter Settings | |
|---|---|---|---|
| Component | Architecture | Hyperparameter | Value |
| Local Model for Accelerometer Data (Client $P^1$) | MLP | Layers | 3 |
| | | Input Size | 348 |
| | | Hidden Size | 140, 70 |
| | | Output Size | 16 |
| | | Activation | ReLU |
| Local Model for Gyroscope Data (Client $P^2$) | MLP | Layers | 3 |
| | | Input Size | 213 |
| | | Hidden Size | 140, 70 |
| | | Output Size | 16 |
| | | Activation | ReLU |
| Global Model | MLP | Layers | 2 |
| | | Input Size | 32 |
| | | Hidden Size | 16 |
| | | Output Size | 6 |
| | | Activation | ReLU |
| Training Params | | Loss | Cross-Entropy |
| | | Epochs | 150 |
| | | Batch Size | 256 |
| | | Optimizer | SGD |
| | | Learning Rate | 0.01 |

predictions based on the concatenation of the embeddings (the final layer produces the logits, without any activation function, and the same setting is applied to other datasets unless specified otherwise).

**KU-HAR** [50]: The KU-HAR dataset is a recent human activity recognition dataset, collected from 90 participants aged 18-34 years. It contains $1,945$ raw activity samples belonging to 18 different classes. The data is collected using smartphone sensors (accelerometer and gyroscope). In addition to the original time-domain samples, the dataset includes $20,750$ subsamples (extracted from the raw data), each containing 3 seconds of data corresponding to a specific activity. The task is to classify the type of activities, and the classification accuracy is reported for the main task performance.

*Dataset Processing:* The data is distributed across two different clients, each with different devices ($K = 2$). The dataset is partitioned into training and test sets with an 8:2 ratio.

*Models:* The details of the models and corresponding parameters are provided in Table 8. For the local models, we use a three-layer MLP to extract the original features into 30-dimensional embeddings. The concatenated embeddings are then sent to the active client, which uses a two-layer MLP as the global model for the final prediction.

## B.2  Robotics

**MuJoCo** [27]: The MuJoCo dataset is used for planar pushing tasks and is collected using MuJoCo [52]. It includes $1,000$ trajectories, each with 250 steps at 10 Hz, simulating the Franka Panda robot arm pushing a circular puck. The pushing actions are generated by a heuristic controller that aims to move the end-effector to the center of the object. The dataset includes data from three different devices: grayscale images of size $32 \times 32 \times 1$ from an RGB camera, forces and binary contact information from sensors, and the 3D position of the end-effector. Additionally, the control inputs are also recorded. The task is to estimate the 2D position of the unknown object on the table surface while the robot interacts with the object. The mean square error of predictions are reported.

*Dataset Processing:* The data from the four devices is distributed across four different clients, i.e., $K = 4$. The dataset contains $1,000$ training records, 10 validation records, and 300 test records. Each data record is split into 29 time-series sequences, each of length 16. The total number of training, validation, and test samples is $29,000$, $290$, and $8,700$, respectively, for a total of $37,990$ samples.

Table 8: **Models and Hyperparemeters for KU-HAR Dataset.**

| Module Description | | Hyperparameter Settings | |
|---|---|---|---|
| Component | Architecture | Hyperparameter | Value |
| Local Model for Accelerometer Data (Client $P^1$) | MLP | Layers
Input Size
Hidden Size
Output Size
Activation | 3
900
300, 100
30
ReLU |
| Local Model for Gyroscope Data (Client $P^2$) | MLP | Layers
Input Size
Hidden Size
Output Size
Activation | 3
900
300, 100
30
ReLU |
| Global Model | MLP | Layers
Input Size
Hidden Size
Output Size
Activation | 2
60
30
18
ReLU |
| Training Params | | Loss
Epochs
Batch Size
Optimizer
Learning Rate | Cross-Entropy
150
256
SGD
0.01 |

*Models:* The details of the models and their parameters for the MuJoCo dataset are provided in Table 9. We follow the settings in [27] for the different devices. For the client with image data, a CNN model combined with an LSTM model is used to extract embeddings. For the clients with sensor data, position data, or control inputs, an MLP and an LSTM model are used to extract embeddings. For the global model in the active client, a single linear model is utilized to make the final prediction.

**VISION&TOUCH [28]:** It is collected from real-world robot simulations, where a 7-DoF torque-controlled Franka Panda robot is used. The simulation involves controlling the robot to perform a peg insertion task. Data is collected using three different sensor devices: proprioception, an RGB-D camera, and a force-torque sensor. The proprioceptive input includes the end-effector pose, as well as linear and angular velocity. The RGB-D camera, positioned to observe the robot, records RGB and depth images, which are then downsampled to a resolution of $128 \times 128$. The force-torque sensor provides 6-axis feedback on forces and moments. In addition, robot action data is recorded at every timestep. The task is to predict the contact state (binary classification, contact or non-contact), and the classification accuracy is reported.

*Dataset Processing:* The four types of sensor data, along with the action data, are distributed across five different clients, i.e., $K = 5$. The dataset contains $147,000$ samples in total, split into training and test sets with an $8 : 2$ ratio.

*Models:* Model details and hyperparameters are provided in Table 10, following the settings in [28] for each type of data. For clients with RGB, depth images, and force-sensor data, a CNN model is used to extract embeddings. For clients with proprioception and action data, an MLP model is applied. Before the global prediction, a sensor fusion strategy as described in [28] is employed, followed by an MLP model in the active client for the final prediction.

## B.3 Healthcare

**MIMIC-III [24]:** This is a large, publicly available database containing de-identified health-related data from over $40,000$ patients admitted to the critical care units of the Beth Israel Deaconess Medical Center between 2001 and 2012. As described in [41], the data for each patient consists of two types: (1) Static information: a set of demographic and personal details such as age, gender, and ethnicity, represented as a 5-dimensional vector; and (2) Time-series information: medical measurements taken hourly over a 24-hour period, with each measurement represented as a 12-dimensional vector

Table 9: **Models and Hyperparemeters for MuJoCo Dataset.**

| Module Description | | Hyperparameter Settings | |
|---|---|---|---|
| Component | Architecture | Hyperparameter | Value |
| Local Model for Position Data (Client $P^1$) | MLP | Layers | 3 |
| | | Input Size | 3 |
| | | Hidden Size | 64, 64 |
| | | Output Size | 64 |
| | | Activation | ReLU |
| | LSTM | Layers | 1 |
| | | Input Size | 64 |
| | | Output Size | 256 |
| Local Model for Image Data (Client $P^2$) | CNN | Layers | 5 |
| | | Kernel Size | 5, 3, 3, 3, 3 |
| | | Num Filters | 32, 32, 32, 16, 8 |
| | | Strides | 1 |
| | | Padding | 2, 1, 1, 1, 1 |
| | LSTM | Layers | 1 |
| | | Input Size | 64 |
| | | Output Size | 256 |
| Local Model for Sensor Data (Client $P^3$) | MLP | Layers | 3 |
| | | Input Size | 7 |
| | | Hidden Size | 64, 64 |
| | | Output Size | 64 |
| | | Activation | ReLU |
| | LSTM | Layers | 1 |
| | | Input Size | 64 |
| | | Output Size | 256 |
| Local Model for Control Data (Client $P^4$) | MLP | Layers | 3 |
| | | Input Size | 7 |
| | | Hidden Size | 64, 64 |
| | | Output Size | 64 |
| | | Activation | ReLU |
| | LSTM | Layers | 1 |
| | | Input Size | 64 |
| | | Output Size | 256 |
| Global Model | Linear Layer | Input Size | 1024 |
| | | Output Size | 6 |
| Training Params | | Loss | MSE |
| | | Epochs | 20 |
| | | Batch Size | 32 |
| | | Optimizer | SGD |
| | | Learning Rate | 0.01 |

(corresponding to 12 different clinical variables). The task is to perform binary classification to determine whether a patient has any ICD-9 code from group 7. The classification accuracy is recorded.

*Data Processing:* The two types of data are assigned to two different clients. The dataset is split into training, validation, and test sets in a ratio of $8:1:1$, which we follow the settings in [33], resulting in $28,970$ training samples, $3,621$ validation samples, and $3,621$ test samples, for a total of $36,212$ samples.

*Models:* Following the setup in [41], we use a two-layer MLP to extract embeddings from the static information and a GRU model to process the time-series data. The hidden states at each time step are recorded and flattened for downstream computation. A two-layer MLP is used as the global model for the final prediction. Details are provided in Table 11.

**PTB-XL [55]:** It is a large-scale electrocardiography (ECG) dataset, comprising $21,700$ clinical 12-lead ECG recordings from $18,885$ patients, designed for a multi-label classification task. There

Table 10: **Models and Hyperparemeters for VISION&TOUCH Dataset.**

| Module Description | | Hyperparameter Settings | |
|---|---|---|---|
| Component | Architecture | Hyperparameter | Value |
| Local Model for RGB Images (Client $P^1$) | CNN | Layers | 6 |
| | | Kernel Size | 7, 5, 5, 3, 3, 3 |
| | | Num Filters | 16, 32, 64, 64, 128, 128 |
| | | Strides | 2 |
| | | Padding | 3,2,2,1,1,1 |
| Local Model for Depth Images (Client $P^2$) | CNN | Layers | 6 |
| | | Kernel Size | 3, 3, 4, 3, 3, 3 |
| | | Num Filters | 32, 64, 64, 64, 128, 128 |
| | | Strides | 2 |
| | | Padding | 1 |
| Local Model for Sensor Data (Client $P^3$) | CNN | Layers | 5 |
| | | Kernel Size | 2 |
| | | Num Filters | 16, 32, 64, 128, 256 |
| | | Strides | 2 |
| | | Padding | 1 |
| Local Model for Proprioception Data (Client $P^4$) | MLP | Layers | 4 |
| | | Input Size | 8 |
| | | Hidden Size | 32, 64, 128 |
| | | Output Size | 256 |
| | | Activation | LeakyReLU |
| Local Model for Action Data (Client $P^5$) | MLP | Layers | 2 |
| | | Input Size | 4 |
| | | Hidden Size | 32 |
| | | Output Size | 32 |
| | | Activation | LeakyReLU |
| Global Model | MLP | Layers | 2 |
| | | Input Size | 200 |
| | | Hidden Size | 128 |
| | | Output Size | 4 |
| | | Activation | LeakyReLU |
| Training Params | | Loss | Cross-Entropy |
| | | Epochs | 15 |
| | | Batch Size | 32 |
| | | Optimizer | SGD |
| | | Learning Rate | 0.001 |

are five diagnostic classes: normal ECG, myocardial infarction, ST/T change, conduction disturbance, and hypertrophy. Following [51], we use the ECG data sampled at 100 Hz. The task is to predict the ECG signal class, where each sample may be associated with multiple labels. For the main task performance, the F1-score is reported.

*Data Processing:* Based on the setup in [1], the time-series readings from different electrodes are divided as follows: (1) limb leads—channels $I$, $II$, $III$, $aVL$, $aVR$, $aVF$, and (2) chest leads—channels $V1$ to $V6$. These are treated as two separate data types. In addition, static patient information such as age and gender (represented as a 4-dimensional vector) is also included as a third data type. These three data types are distributed across three different clients. The dataset is divided into training, validation, and test sets in a ratio of $8 : 1 : 1$.

*Models:* We adopt a similar setup to the MIMIC-III dataset, where GRU models are used to extract local embeddings and an MLP model is used for the final prediction. Details are provided in Table 12.

Table 11: **Models and Hyperparemeters for MIMIC-III Datasets.**

| Module Description | | Hyperparameter Settings | |
|---|---|---|---|
| *Component* | *Architecture* | *Hyperparameter* | *Value* |
| Local Model for Static Information Data (Client $P^1$) | MLP | Layers | 2 |
| | | Input Size | 5 |
| | | Hidden Size | 10 |
| | | Output Size | 10 |
| | | Activation | LeakyReLU |
| Local Model for Time-Series Data (Client $P^2$) | GRU | Layers | 1 |
| | | Input Size | 12 |
| | | Output Size | 30 |
| Global Model | MLP | Layers | 2 |
| | | Input Size | 730 |
| | | Hidden Size | 40 |
| | | Output Size | 2 |
| | | Activation | LeakyReLU |
| Training Params | | Loss | Cross-Entropy |
| | | Epochs | 300 |
| | | Batch Size | 256 |
| | | Optimizer | SGD |
| | | Learning Rate | 0.01 |

Table 12: **Models and Hyperparemeters for PTB-XL Datasets.**

| Module Description | | Hyperparameter Settings | |
|---|---|---|---|
| *Component* | *Architecture* | *Hyperparameter* | *Value* |
| Local Model for Static Information Data (Client $P^1$) | MLP | Layers | 2 |
| | | Input Size | 4 |
| | | Hidden Size | 10 |
| | | Output Size | 10 |
| | | Activation | LeakyReLU |
| Local Model for Limb Lead Data (Client $P^2$) | GRU | Layers | 1 |
| | | Input Size | 6 |
| | | Output Size | 15 |
| Local Model for Chest Lead Data (Client $P^3$) | GRU | Layers | 1 |
| | | Input Size | 6 |
| | | Output Size | 15 |
| Global Model | MLP | Layers | 2 |
| | | Input Size | 30010 |
| | | Hidden Size | 100 |
| | | Output Size | 5 |
| | | Activation | LeakyReLU |
| Training Params | | Loss | BCEWithLogits |
| | | Epochs | 100 |
| | | Batch Size | 256 |
| | | Optimizer | SGD |
| | | Learning Rate | 0.01 |

## B.4 Emotion Analysis

**UR-FUNNY [20]:** It is the first large-scale humor detection collection that incorporates image, text, and audio data. It contains a total of $16,514$ video segments sourced from $1,866$ videos, along with their transcripts obtained from the TED portal. Each video is labeled as either humorous or non-humorous, and the task involves predicting whether a given video segment contains humor. For the main task performance, the classification accuracy is recorded.

Table 13: **Models and Hyperparemeters for UR-FUNNY and MUSTARD Datasets.**

| Module Description | | Hyperparameter Settings | |
|---|---|---|---|
| Component | Architecture | Hyperparameter | Value |
| Local Model for Video Features (Client $P^1$) | GRU | Layers | 1 |
| | | Input Size | 371 |
| | | Output Size | 700 |
| Local Model for Audio Features (Client $P^2$) | GRU | Layers | 1 |
| | | Input Size | 81 |
| | | Output Size | 160 |
| Local Model for Text Features (Client $P^3$) | GRU | Layers | 1 |
| | | Input Size | 300 |
| | | Output Size | 600 |
| Global Model | MLP | Layers | 2 |
| | | Input Size | 1460 |
| | | Hidden Size | 1460 |
| | | Output Size | 2 |
| | | Activation | LeakyReLU |
| Training Params | | Loss | Cross-Entropy |
| | | Epochs | 100 |
| | | Batch Size | 256 |
| | | Optimizer | SGD |
| | | Learning Rate | 0.01 |

Table 14: **Models and Hyperparemeters for CMU-MOSI and CMU-MOSEI Datasets.**

| Module Description | | Hyperparameter Settings | |
|---|---|---|---|
| Component | Architecture | Hyperparameter | Value |
| Local Model for Video Features (Client $P^1$) | GRU | Layers | 1 |
| | | Input Size | 35 |
| | | Output Size | 70 |
| Local Model for Audio Features (Client $P^2$) | GRU | Layers | 1 |
| | | Input Size | 74 |
| | | Output Size | 200 |
| Local Model for Text Features (Client $P^3$) | GRU | Layers | 1 |
| | | Input Size | 300 |
| | | Output Size | 600 |
| Global Model | MLP | Layers | 2 |
| | | Input Size | 870 |
| | | Hidden Size | 870 |
| | | Output Size | 1 |
| | | Activation | LeakyReLU |
| Training Params | | Loss | MSE |
| | | Epochs | 100 |
| | | Batch Size | 256 |
| | | Optimizer | SGD |
| | | Learning Rate | 0.05 |

*Dataset Processing:* Following the settings in [20], we extract sequential features from the original data for video, text, and audio. The extracted feature dimensions are $T \times 371$ for video, $T \times 300$ for text, and $T \times 81$ for audio, where $T$ represents the sequence length ranging from 5-20 seconds. These three data types are then distributed across three different clients. The dataset is partitioned into training ($10,598$ segments), validation ($2,626$ segments), and test sets ($3,290$ segments).

*Models:* For local models, we employ the widely-used models in time-series data, GRU models [11], to extract embeddings from video, audio, and text features. The hidden state from the final time step

is utilized for subsequent computations. In the global model, we implement a two-layer MLP for the final prediction. Detailed model parameters are provided in Table 13.

**MUSTARD [7]:** It is designed for sarcasm detection and consists of video segments collected from web searches, primarily from YouTube. The videos are sourced from four popular TV shows: 'Friends', 'The Golden Girls', 'Sarcasmaholics Anonymous', and 'The Big Bang Theory'. The dataset includes 690 audio-video segment pairs, along with contextual information to aid classification. Each sample is labeled for binary classification as either sarcastic or non-sarcastic. The classification accuracy is reported for the main task performance.

*Dataset Processing:* Following the same methodology as [20], we extract features from the original video, text, and audio data, with dimensions of $T \times 371$, $T \times 300$, and $T \times 81$, respectively, where $T$ denotes the sequence length. These three data types are distributed across three distinct clients. The dataset is partitioned into training, validation, and test sets in a $6 : 2 : 2$ ratio.

*Models:* We employ the same model architecture as used for the UR-FUNNY dataset. For detailed parameters, refer to Table 13.

**CMU-MOSI [70]:** It is the first opinion-level annotated corpus of sentiment analysis in online videos. It consists of $2,199$ opinionated video clips, each labeled with sentiment intensity on a $[-3, 3]$ scale: $[-3$: highly negative, $-2$: negative, $-1$: weakly negative, $0$: neutral, $+1$: weakly positive, $+2$: positive, $+3$: highly positive]. The dataset is collected from YouTube, focusing on video blogs (vlogs) that reflect real-world speaker behaviors in monologue videos. For our experiments, we adopt a binary classification task, dividing sentiment into positive ($> 0$) or negative ($< 0$), a common setting in emotion analysis. The classification accuracy is reported for the main task performance.

*Dataset Processing:* Following existing work in emotion analysis [70], we extract feature sequences for image, text, and audio data from the original dataset, with the sizes of $T * 35$, $T * 300$, and $T * 74$, where $T$ is the length of each sequence. The data is distributed across three different clients by data type. The dataset is split into training, validation, and test sets with $1,284$, $229$, and $686$ clips.

*Models:* We use GRU models to extract embeddings, and a two-layer MLP for final prediction. The models are trained using the MSE loss. During testing, the final predictions are converted into two classes as described above. Detailed model parameters are presented in Table 14.

**CMU-MOSEI [71]:** It is a large-scale sentiment and emotion analysis dataset from real-world online videos. It contains 22,777 data samples with labels for sentiment intensity on the scale of $[-3, 3]$. We use the binary classification task that divides the sentiment as positive ($> 0$) or negative ($< 0$). The classification accuracy is recorded.

*Data Processing:* We utilize the settings of existing works [70, 71] to extract features from the original data, with sizes of $T \times 35$, $T \times 300$, and $T \times 74$ for image, text, and audio features, where $T$ is the length of each sequence. The data is distributed into three different clients by different data types. The dataset is split into training, validation, and test sets with $16,265$, $1,869$, and $4,643$ segments, respectively.

*Models:* We use the same settings as the CMU-MOSEI dataset, detailed in Table 14.

### B.5 Multimedia

**NUS-WIDE [10]:** It is a dataset collected from the Flickr website with associated tags. The dataset is preprocessed into $634$ dimensions of image features and $1,000$ dimensions of text features. The task is to classify $81$ concept types using the input image and text features. The classification accuracy is utilized for comparing the main task performance.

*Dataset Processing:* We use a five-class subset [37] including 'buildings', 'grass', 'animal', 'water', and 'person', with $69,966$ samples in the training set and $46,693$ samples in the test set. The image features and the text features are then distributed to two clients, respectively.

*Models:* For both the image and text data, we utilize a three-layer MLP to extract embeddings from the features, and a two-layer MLP is utilized to make the final prediction. Detailed in Table 15.

**MM-IMDB [3]:** It is a large-scale dataset for movie genre prediction, developed based on the Movielens dataset [19]. It expands the dataset by collecting movie genres, posters (images), and plot information (text) for each movie. It is built using the IMDb ids provided by the Movielens dataset,

Table 15: **Models and Hyperparemeters for NUS-WIDE Dataset.**

| Module Description | | Hyperparameter Settings | |
|---|---|---|---|
| Component | Architecture | Hyperparameter | Value |
| Local Model for Image Features (Client $P^1$) | MLP | Layers | 3 |
| | | Input Size | 634 |
| | | Hidden Size | 320, 80 |
| | | Output Size | 40 |
| | | Activation | ReLU |
| Local Model for Text Features (Client $P^2$) | MLP | Layers | 3 |
| | | Input Size | 1000 |
| | | Hidden Size | 500, 125 |
| | | Output Size | 60 |
| | | Activation | ReLU |
| Global Model | MLP | Layers | 2 |
| | | Input Size | 100 |
| | | Hidden Size | 50 |
| | | Output Size | 5 |
| | | Activation | ReLU |
| Training Params | | Loss | Cross-Entropy |
| | | Epochs | 30 |
| | | Batch Size | 256 |
| | | Optimizer | $SGD$ |
| | | Learning Rate | 0.02 |

Table 16: **Models and hyper-params for MM-IMDB dataset.**

| Module Description | | Hyperparameter Settings | |
|---|---|---|---|
| Component | Architecture | Hyperparameter | Value |
| Local Model for Image Features (Client $P^1$) | MaxoutMLP | Layers | 2 |
| | | Input Size | 300 |
| | | Hidden Size | 512 |
| | | Output Size | 512 |
| Local Model for Text Features (Client $P^2$) | MaxoutMLP | Layers | 3 |
| | | Input Size | 4096 |
| | | Hidden Size | 1024, 512 |
| | | Output Size | 512 |
| Global Model | Linear Layer | Input Size | 1024 |
| | | Output Size | 23 |
| Training Params | | Loss | Cross-Entropy |
| | | Epochs | 100 |
| | | Batch Size | 256 |
| | | Optimizer | SGD |
| | | Learning Rate | 0.005 |

which contains ratings for $27,000$ movies. Movies without poster images were filtered out, resulting in a final dataset with ratings for $25,959$ movies. The task is to predict the class of movie genres, which may involve multiple labels. The F1-score is used for comparing the main task performance.

*Dataset Processing:* We follow the settings of [3] to extract features from images and texts. The final image and text features have dimensions of 300 and $4,096$, respectively. The image and text features are distributed to two clients. The samples are split into training, validation, and test sets with $15,552$, $2,608$, and $7,799$ samples, respectively.

*Models:* Following the settings in [3], we use the MaxoutMLP architecture [17] with the same hyper-parameters. For the global model, we use a two-layer MLP to make the final prediction, as detailed in Table 16.

# C Evaluation Setting

This section provides a comprehensive overview of the evaluation settings employed in our experiments. All experiments were conducted on a server with 8 NVIDIA GeForce RTX 4090 GPUs, equipped with Intel(R) Xeon(R) Gold 6240 CPUs operating at 2.60 GHz.

## C.1 Efficiency

**Evaluation Metrics.** MARS-VFL provides three different evaluation metrics to evaluate the main task performance (MP), ranging from classification accuracy, F1-score, and mean squared error (MSE) for different prediction tasks:

(1) *Classification Accuracy ($\mathcal{A}$):* Given the VFL model $f(\Theta)$ and the test dataset $\{(x_i, y_i)\}_{i=1}^{N_t}$, for classification tasks, the classification accuracy $\mathcal{A}$ is reported:

$$\mathcal{A} = \frac{1}{N_t} \sum_{i=1}^{N_t} \mathbb{I}\left(f(x_i, \Theta) = y_i\right), \tag{7}$$

where $\mathbb{I}(\cdot)$ is the indicator function that returns 1 if the condition is true and 0 otherwise.

(2) *F1-score ($F_1$):* For the multi-label classification tasks, we utilize the F1-score to evaluate the performance:

$$F_1 = 2 \cdot \frac{P \cdot R}{P + R}, \quad P = \frac{TP}{TP + FP}, \quad R = \frac{TP}{TP + FN}, \tag{8}$$

where $TP$, $FP$, and $FN$ represent the number of true positives, false positives, and false negatives in predictions, respectively.

(3) *Mean Squared Error (MSE):* For the regression tasks, we use the MSE as the evaluation metric. We report the MP of the test set on each dataset:

$$\text{MSE} = \frac{1}{N_t} \sum_{i=1}^{N_t} \|f(x_i, \Theta) - y_i\|_2^2, \tag{9}$$

where $\| \cdot \|_2$ denotes the Euclidean norm. Besides, the communication costs as well as the training epochs are also reported:

(4) *Communication Costs:* We report the communication costs of exchanged information during the training stage. Communication costs account for both the transmission of embeddings between active and passive clients and the transmission of gradients from the active client to the passive clients. The units frequently used are MB and GB, based on the scales of different datasets.

(5) *Epochs:* We report the training epochs required to reach the maximum test accuracy, and we also represent the valid/test MP and communication costs with respect to epochs, which also reflects the convergence speed of different methods.

**Evaluation Protocols.** MARS-VFL evaluates the efficiency of different methods by running each experiment five times and reporting the mean values and standard deviations from three aspects: (1) Report the maximum MP, the corresponding communication costs, and epochs. This not only compares the performance of different methods but also their efficiency and convergence epochs. (2) Compare the MP under equal communication cost to assess the communication efficiency of each method. It provides more qualitative results for comparing the efficiency of different methods. (3) Compare the MP under the same number of epochs. It shows the performance and convergence behavior across epochs.

**Evaluated Methods.** We evaluate three recent methods: (1) FedBCD [36] aims to reduce communication costs and accelerate convergence speed through local updates. It uses historical gradients to update local models in multiple rounds before global updates, achieving lower costs and fewer global epochs. (2) Based on local updates, C-VFL [6] proposes compressing the intermediate embedding to achieve low costs. The communication costs are reduced based on the compression rates. However, compression can lead to information loss during training. (3) EFVFL [54] introduces an error feedback mechanism to mitigate information loss caused by compression during training, especially at small compression rates. It enables the maintenance of model performance while significantly

reducing communication overhead. We employ local updates with EFVFL to align the experiment settings, and the local update rounds are set to $5$ for all three methods. For C-VFL and EFVFL, the compression rates are set to $0.3$ for all experiments.

## C.2 Robustness

**Evaluation Metrics.** To evaluate the robustness of different methods, MARS-VFL assesses the main task performance (the same as efficiency) under different perturbation rates. We report the MP under different training/validation set perturbation rates $r_a$ and test set perturbation rates $r_b$.

**Evaluation Protocols.** We evaluate the robustness of different methods with different perturbation rates. (1) For missing features, following existing settings [53], the perturbation rates refer to the probability of different feature parts of samples being missing, and the MP is reported for different missing probabilities. We test missing probabilities in $\{0, 0.2, 0.5, 0.8\}$. (2) For corrupted features, the perturbation rates refer to the proportion of corrupted feature parts, which is practical in scenarios where parts of features might be corrupted in each sample. The MP is reported for different corruption rates. We use Gaussian noise to simulate data corruption, which can be extended to other corruption types. Gaussian noise with standard deviations in $\{0.1, 0.2, 0.4, 0.6, 0.8\}$ is randomly sampled and applied to randomly selected feature parts according to the corruption rates. We test corruption rates in $\{0, 0.2, 0.5, 0.8\}$. (3) For misaligned features, the perturbation rates refer to the proportion of shuffled samples. We randomly shuffle the samples with ratios in $\{0, 0.5, 0.8, 1\}$, and the MP is reported for different shuffling ratios.

**Evaluated Methods.** We evaluate several methods for different perturbations:

*(1) Missing Features:* We evaluate LEEF-VFL [47] and LASER-VFL [53]. (1) LEEF-VFL proposes performing local updates before each collaboration round by leveraging private labels. It fully utilizes all training samples and information from private labels, enhancing robustness against missing features. (2) LASER-VFL proposes leveraging different subsets of features and training predictors with mean embedding aggregation, achieving good performance across different combinations of feature blocks. Following the settings in [53], since the standard VFL model can only be trained on and used for inference with fully observed samples, for Base methods, samples with missing features are ignored in both training and testing, and a random test prediction is made. For LEEF-VFL, samples with missing features can be utilized for training, but are ignored, and a random prediction is made during testing. For LASER-VFL, the use of mean aggregation for embeddings and parameter sharing across multiple predictors enables both training and testing to be conducted with missing features. However, it can only be generalized to settings with the same embedding sizes for each client; in other settings, only the evaluations of Base and LEEF-VFL are conducted.

*(2) Corrupted Features:* We propose RVFL-Aug, which utilizes feature augmentations to learn inherent and consistent information from the samples, thereby enhancing robustness against data corruption. We utilize three different augmentations: (a) random mask, which randomly sets parts of the features to zero; (b) random scale up, which randomly multiplies parts of the features by $1.2$; (c) random scale down, which randomly multiplies parts of the features by $0.8$. For the strength of the JS consistency constraint, we adjust $\lambda$ in $\{1, 3, 6, 12\}$ and report the best performances.

*(3) Misaligned Features:* We propose RVFL-Align, which maximizes embedding consistency between sample features from different parties, thereby aligning them optimally. It is a training-free process, and we utilize it in each forward process, achieving promising results in resisting misaligned features.

## C.3 Security

**Evaluation Metrics.** In MARS-VFL, the evaluation of attack performance (AP) is conducted using three well-established metrics, each designed to capture a specific dimension of attack effectiveness. These metrics provide a rigorous and comprehensive assessment of the vulnerabilities in VFL systems under various threat models.

*(1) Label Inference Performance (LIP):* In label inference attacks, the inference accuracy on the test set is reported. Given the inferred label set $\{\hat{y}_i\}_{i=1}^{N_t}$ and the true label set $\{y_i\}_{i=1}^{N_t}$, the LIP can be

defined as:

$$\text{LIP} = \frac{1}{N_t} \sum_{i=1}^{N_t} \mathbb{I}(\hat{y}_i = y_i), \tag{10}$$

where $\mathbb{I}(\cdot)$ is the indicator function that returns 1 if the condition is true and 0 otherwise.

*(2) Feature Inference Performance (FIP):* In feature inference attacks, following the setting in [38], the mean squared error per feature between the inferred features and the original features is reported. To be consistent with other attack metrics, we use a negative correlation-based metric to evaluate the inference performance. Given the inferred feature set $\{\hat{x}_i\}_{i=1}^{N_t}$ and the original feature set $\{x_i\}_{i=1}^{N_t}$, we normalize the features to $[a, b]$, where $b - a = C$, before performing attacks. The FIP can be defined as:

$$\text{FIP} = \frac{C - \text{MSE}_p}{C}, \quad \text{MSE}_p = \frac{1}{N_t \cdot d} \sum_{i=1}^{N_t} \|\hat{x}_i - x_i\|_2^2, \tag{11}$$

where $\|\cdot\|_2$ denotes the Euclidean norm, $d$ denotes the feature dimensions, and $[a, b]$ denotes the normalized feature range. For instance, to perform attacks on the UCI-HAR and NUS-WIDE datasets, the features of the attack target are normalized to $[-1, 1]$, with $C = 2$.

*(3) Attack Success Rate (ASR):* In backdoor attacks, the attack success rate on poisoned samples is widely used to evaluate the attack performance. Given the target class $\tau$, the poisoned feature set $\{\hat{x}_i\}_{i=1}^{N_p}$ and the poisoned model $f(\hat{\Theta})$, the ASR is defined as follows:

$$\text{ASR} = \frac{1}{N_p} \sum_{i=1}^{N_p} \mathbb{I}(f(\hat{x}_i, \Theta) = \tau). \tag{12}$$

We report the attack performance with different defenses, and we also report the main task performance (AP) to evaluate the influence of different attacks and defenses on the collaboration tasks.

**Evaluation Protocols.** We evaluate the security issues from three aspects: (1) For different attack methods, we evaluate the attack performance (AP) as well as the to show the effectiveness of each method. (2) We evaluate the attack performance under different defense methods to explore potential solutions for malicious attacks. (3) We also provide the main task performance (MP) to assess the influence of different defense methods on the prediction tasks, as well as the influence of attack methods on the prediction tasks.

**Evaluated Methods.** We evaluate different attack methods as well as defense methods:

*Label Inference Attacks:* We evaluate two attack methods, including PMC [15] and AMC [15]. PMC and AMC propose to construct an inference classifier using the trained local model of the attacker. This requires a small set of labeled samples as prior information. Compared with PMC, AMC integrates an additional learning rate adjustment to scale up the influence of the attack model. The learning rates of the PMC and AMC are set to 0.01 for all experiments. For the UCI-HAR dataset, the number of labeled samples per class is set to 20, while for the NUS-WIDE dataset, it is set to 100.

*Feature Inference Attacks:* We evaluate two attack methods, including GRNA [38] and MIA [30]. (1) GRNA infers target features based on the final outputs of the active client, the model parameters, and auxiliary features from the same sample. (2) In contrast, MIA assumes access to part of the original feature values and model parameters during training and learns a model to recover the original features from the active client's embedding. As GRNA lacks access to ground-truth feature values, its reconstruction performance is expected to be inferior to that of MIA; however, GRNA can generalize to more common settings since the original features are usually not available. For all experiments, we set $P^1$ as the attacker client and $P^0$ as the client of the victim. For the UCI-HAR dataset, the learning rates are set to 0.012 for GRNA and 0.008 for MIA. For the NUS-WIDE dataset, the learning rates are set to 0.02 for GRNA and 0.015 for MIA.

*Backdoor Attacks:* We evaluate two attack methods, including TECB [8] and LFBA [46]. (1) TECB injects backdoors by leveraging a small fraction of labeled target class samples and model parameters. It optimizes a backdoor trigger based on the labeled samples and poisons the model during VFL training. It achieves high attack success rates even with minimal target label knowledge. (2) LFBA, on the other hand, operates without access to label information. Instead of a preset poison set, it exploits the information of the embedding gradients during the training stage and constructs the target

sample set for poisoning. For TECB, the learning rates are set to $0.02$ and $0.001$ for the UCI-HAR and NUS-WIDE datasets, respectively, while keeping other settings the same as in [8]. For LFBA, we follow the settings in [46].

*Defenses:* We evaluate four gradient-based defenses following the settings in [15], including Noisy Gradients (NG) [76], Gradient Compression (GC) [35, 25], Privacy-preserving Deep Learning (PPDL) [49], and DiscreteSGD (DSGD) [4, 15]. (1) Noisy Gradients (NG): As described in [76], adding noise to gradients is a common defense strategy in FL. In VFL setting, we apply Laplacian noise of different scales to gradients before transmitting them to clients, as in [15]. (2) Gradient Compression (GC): It mitigates attacks by sharing only a subset of gradients with the highest absolute values. Originally proposed to improve communication efficiency [35], it also offers privacy benefits [25]. We evaluate its effectiveness by applying different compression ratios. (3) Privacy-preserving deep learning (PPDL): Introduced in [49], it combines differential privacy, gradient compression, and random selection. At each iteration, the server (i) randomly selects a gradient value and adds noise; (ii) retains it only if the noisy value exceeds a threshold; and (iii) repeats until a certain fraction of the gradients is preserved. We assess its performance in VFL using different retention rates, following [15]. (4) DiscreteSGD (DSGD): Adapted for VFL from SignSGD [4], this defense observes the gradient distribution during the first epoch to compute the mean $\mu$ and standard deviation $\sigma$. Using the three-sigma rule [40], an interval $[\mu - 2\sigma, \mu + 2\sigma]$ is defined, and values outside it are treated as outliers. The interval is then divided into several sub-intervals, and gradient values are rounded to the nearest endpoint before being shared. We change the number of sub-intervals to control the level of information retained, where a larger interval implies finer granularity and weaker defense.

# D    Extended Analysis

We provide analysis of time complexity and communication costs of RVFL-Aug and RVFL-Align.

## D.1    RVFL-Aug

**Time Complexity.** Let $N$ denote the number of aligned training samples, $K$ as the number of clients, and $S$ as the number of augmentation sequences. The time complexity for processing all samples at client $P^k$ in one epoch is:

$$\text{Time}_k = \underbrace{O(NS)}_{\text{Augmentation}} + \underbrace{O(3N)}_{\text{Forward}} + \underbrace{O(3N)}_{\text{Backward}}. \tag{13}$$

This includes the cost of data augmentation, and the forward and backward passes for three views. The total per-round time complexity across all $K$ clients is:

$$\text{Time}_{\text{per}} = \sum_{k=1}^{K} O(7NS). \tag{14}$$

Over $T$ training epochs, the overall time complexity becomes:

$$\text{Time}_{\text{total}} = \sum_{k=1}^{K} O(T(7NS)). \tag{15}$$

**Communication Costs.** In each training round, client $P^k$ generates and transmits three embeddings corresponding to the original input and two augmented views, denoted as $H^k$, $H_{a_1}^k$, and $H_{a_2}^k$. Assuming the embedding dimensions of $d^k$ and $N$ samples per client, the communication cost per client per round is $3Nd^k$ for both forward and backward transmissions. Thus, the total communication costs per round across all clients is:

$$\text{Comm}_{\text{per}} = \sum_{k=1}^{K} (3Nd^k + 3Nd^k) = \sum_{k=1}^{K} 6Nd^k. \tag{16}$$

Over $T$ training rounds, the total communication costs become:

$$\text{Comm}_{\text{total}} = \sum_{k=1}^{K} 6TNd^k. \tag{17}$$

In summary, RVFL-Aug introduces additional linear time and communication overheads with respect to the sample size and embedding dimension, which remain acceptable in practical scenarios.

## D.2 RVFL-Align

**Time Complexity.** Denote $N$ as the number of aligned training samples, and $K$ as the total number of clients. The main computational costs in RVFL-Align come from two parts:

- *Consistency Matrix Calculation.* For each passive client $P^k$, the active client computes a pairwise cosine similarity matrix $\mathcal{C}^k \in \mathbb{R}^{N \times N}$ between its own embeddings $H^a$ and the embeddings $H^k$ from $P^k$. This step requires $O(N^2)$ time complexity per passive client.
- *Solving the Linear Assignment Problem.* For each similarity matrix $\mathcal{C}^k$, the optimal matching matrix $M^k$ is obtained by solving a linear assignment problem, which takes $O(N^3)$ time using the Hungarian algorithm [26].

Thus, the total time complexity across all $K - 1$ passive clients over $T$ training epochs is:

$$\text{Time}_{\text{total}} = \sum_{k=1}^{K-1} O\big(T(N^2 + N^3)\big), \tag{18}$$

which is dominated by the $O(N^3)$ term when $N$ is large. RVFL-Align introduces additional computational overhead with $O(N^3)$ complexity, which may be resource-intensive in large-scale scenarios.

**Communication Costs.** RVFL-Align does not introduce additional communication overhead beyond standard VFL. Each passive client $P^k$ sends its embeddings $H^k \in \mathbb{R}^{N \times d^k}$ to the active client $P^a$ for consistency evaluation, and receives the corresponding embedding gradients in return. Hence, the communication cost per round from all passive clients is:

$$\text{Comm}_{\text{per}} = \sum_{k=1}^{K-1} ( \underbrace{N d^k}_{\text{Forward}} + \underbrace{N d^k}_{\text{Backward}} ). \tag{19}$$

Over $T$ training rounds, the total communication cost is:

$$\text{Comm}_{\text{total}} = \sum_{k=1}^{K-1} 2TN d^k. \tag{20}$$

In summary, RVFL-Align may face scalability challenges due to its additional time complexity, but it provides an effective approach for handling feature misalignment. Future work can explore more efficient algorithms to improve its scalability.

# E   Additional Results

Due to space limitations in the main text, the remaining results are reported in the appendix, following the same experiment settings.

## E.1 Efficiency

We present additional efficiency evaluation results on the VISION&TOUCH and MuJoCo datasets in Table 17, where MP refers to classification accuracy $\mathcal{A}(\%)$ or mean squared error (MSE), respectively. As shown in Table 17, compared to the Base method, FedBCD, C-VFL, and EFVFL achieve lower communication costs and faster convergence, but at the expense of main task performance, raising *the trade-off between main task performance and communication efficiency*, as well as *between main task performance and convergence speed*. Furthermore, although C-VFL and EFVFL achieve reduced communication costs through embedding compression compared to FedBCD, they require more training epochs to converge, indicating *the trade-off between communication cost and convergence speed*. Investigating such trade-offs is valuable for developing more efficient VFL systems.

Table 17: **Results of Efficiency.** The results on the VISION&TOUCH and MuJoCo are reported.

| Method | VISION&TOUCH | | | MuJoCo | | |
|---|---|---|---|---|---|---|
| | Best MP (%) | Costs (GB) | Epochs | Best MP (MSE) | Costs (GB) | Epochs |
| Base | **92.18±0.64** | 3.41±1.28 | 8±3 | **1.52±0.13** | 29.21±13.27 | 11±5 |
| FedBCD [36] | 91.76±0.42 | 0.85±0.42 | **2±1** | 1.73±0.24 | 10.62±5.31 | **4±2** |
| C-VFL [6] | 91.49±0.57 | **0.38±0.13** | 3±2 | 1.76±0.12 | **3.98±2.39** | 5±3 |
| EFVFL [54] | 91.83±0.61 | 0.51±0.25 | 4±2 | 1.74±0.25 | 5.58±1.59 | 7±2 |

## E.2  Robustness

We provide full results of robustness evaluations in Table 18, Table 19 and Table 20.

**Missing Features.** We provide the complete evaluation results under missing feature scenarios in Table 18. As shown in Table 18, both LEEF-VFL and LASER-VFL demonstrate strong performance under different missing rates. However, LASER-VFL outperforms LEEF-VFL as the missing rate increases in the test sets, owing to the utilization of mean aggregation for embeddings and parameter sharing across multiple predictors tailored to different feature combinations. Despite its advantage in handling missing testing features, LASER-VFL is limited to cases where the embedding dimensions across clients are equal, due to its reliance on mean aggregation. For settings with unequal embedding dimensions, the results of LEEF-VFL are reported.

**Corrupted Features.** We report the results under corrupted feature settings in Table 19. As shown in the results, the proposed RVFL-Aug demonstrates strong robustness against various corruption rates, particularly under high corruption rates, highlighting the effectiveness of consistency learning through augmentations. However, performance degradation is observed in some settings where features are not corrupted. For example, on the UCI-HAR dataset with $r_a = 0$ and $r_b = 0$, RVFL-Aug leads to a drop in performance. This may be attributed to additional noise introduced by the augmentations. It is valuable to explore methods that are resilient to both clean and corrupted data in future work.

**Misaligned Features.** We present the complete results under misaligned feature settings in Table 20. The proposed RVFL-Align demonstrates strong robustness against different rates of misaligned samples, highlighting the effectiveness of realigning embeddings through maximum consistency. However, its performance can degrade compared to Base when the original features are well aligned. For example, on the UCI-HAR dataset with $r_a = 0$ and $r_b = 0$, where some aligned samples may be unnecessarily shuffled by RVFL-Align. In addition, RVFL-Align introduces extra computational complexity (as detailed in Section D.2). For future work, it would be valuable to explore more efficient methods that maintain strong performance both under well-aligned and misaligned settings.

## E.3  Security

We present comprehensive security evaluation results on the UCI-HAR and NUS-WIDE datasets in Table 21 and Table 22. The results demonstrate that VFL systems are highly vulnerable to various types of attacks. Specifically, label inference attacks achieve over 60% inference accuracy, feature inference attacks exceed 90% average precision (AP), and backdoor attacks attain success rates surpassing 99%. These findings highlight the critical security risks associated with deploying VFL in real-world scenarios. While some defense mechanisms have been proposed, they remain insufficient to fully mitigate these threats. In particular, although certain defenses can reduce the effectiveness of attacks, they often do so at the expense of significantly degrading the performance of the learning task. This trade-off underscores the urgent need for more robust and adaptive defense strategies. Moreover, as demonstrated by the results, gradient-based defenses are insufficient against attacks that do not rely on gradient information, such as MIA (an embedding-based attack), thereby presenting challenges in developing effective methods to mitigate such threats. Future research should focus on designing security mechanisms that not only counteract a wide spectrum of attacks but also preserve the utility and accuracy of VFL systems in practical deployments.

Table 18: **Results with Missing Features.**

### UCI-HAR (MP: $\mathcal{A}$ (%))

| Method | $r_a = 0$ | | | | $r_a = 0.2$ | | | | $r_a = 0.5$ | | | | $r_a = 0.8$ | | | |
|---|---|---|---|---|---|---|---|---|---|---|---|---|---|---|---|---|
| | $r_b=0$ | $r_b=0.2$ | $r_b=0.5$ | $r_b=0.8$ | $r_b=0$ | $r_b=0.2$ | $r_b=0.5$ | $r_b=0.8$ | $r_b=0$ | $r_b=0.2$ | $r_b=0.5$ | $r_b=0.8$ | $r_b=0$ | $r_b=0.2$ | $r_b=0.5$ | $r_b=0.8$ |
| Base | 94.67 | 90.77 | 90.50 | 56.67 | 94.47 | 90.23 | 89.55 | 55.99 | 93.93 | 89.65 | 89.48 | 56.26 | 74.65 | 69.56 | 64.00 | 45.74 |
| LEEF-VFL [47] | 94.60 | 90.80 | 91.92 | 57.28 | 95.05 | 91.25 | 91.04 | 57.48 | 95.76 | 90.70 | 90.22 | 57.99 | 91.38 | 87.34 | 81.13 | 55.17 |
| LASER-VFL [53] | **95.88** | **94.33** | **95.37** | **76.05** | **95.64** | **95.01** | **95.32** | **76.13** | 95.76 | **94.91** | **94.79** | **76.44** | **91.48** | **88.92** | **90.79** | **71.38** |

### KU-HAR (MP: $\mathcal{A}$ (%))

| Method | $r_a = 0$ | | | | $r_a = 0.2$ | | | | $r_a = 0.5$ | | | | $r_a = 0.8$ | | | |
|---|---|---|---|---|---|---|---|---|---|---|---|---|---|---|---|---|
| | $r_b=0$ | $r_b=0.2$ | $r_b=0.5$ | $r_b=0.8$ | $r_b=0$ | $r_b=0.2$ | $r_b=0.5$ | $r_b=0.8$ | $r_b=0$ | $r_b=0.2$ | $r_b=0.5$ | $r_b=0.8$ | $r_b=0$ | $r_b=0.2$ | $r_b=0.5$ | $r_b=0.8$ |
| Base | 82.24 | 66.41 | 38.12 | 23.98 | 81.49 | 65.93 | 38.48 | 23.71 | 79.98 | 61.78 | 37.61 | 23.16 | 73.47 | 58.46 | 33.20 | 21.57 |
| LEEF-VFL [47] | 82.22 | **75.54** | 43.81 | 34.58 | 81.59 | 74.00 | 43.18 | 34.14 | 80.41 | 72.39 | 42.12 | 33.37 | 73.46 | 66.00 | 33.42 | 26.96 |
| LASER-VFL [53] | **82.69** | 75.42 | **65.03** | **79.20** | **81.83** | **74.82** | **63.38** | **80.76** | **80.55** | **74.57** | **64.06** | **75.98** | **75.42** | **68.86** | **58.21** | **71.09** |

### MM-IMDB (MP: $F_1$ (%))

| Method | $r_a = 0$ | | | | $r_a = 0.2$ | | | | $r_a = 0.5$ | | | | $r_a = 0.8$ | | | |
|---|---|---|---|---|---|---|---|---|---|---|---|---|---|---|---|---|
| | $r_b=0$ | $r_b=0.2$ | $r_b=0.5$ | $r_b=0.8$ | $r_b=0$ | $r_b=0.2$ | $r_b=0.5$ | $r_b=0.8$ | $r_b=0$ | $r_b=0.2$ | $r_b=0.5$ | $r_b=0.8$ | $r_b=0$ | $r_b=0.2$ | $r_b=0.5$ | $r_b=0.8$ |
| Base | 56.35 | 31.79 | 19.99 | 18.80 | 55.26 | 30.43 | 19.88 | 18.50 | 54.95 | 30.42 | 19.89 | 18.33 | 23.21 | 21.85 | 18.63 | 18.21 |
| LEEF-VFL [47] | 56.53 | 35.77 | 21.22 | 18.24 | 55.29 | 36.22 | 21.82 | 18.19 | 55.04 | 35.33 | 21.35 | 18.08 | 20.61 | 19.99 | 18.73 | 17.72 |
| LASER-VFL [53] | **56.62** | **50.44** | **50.30** | **50.43** | **55.47** | **50.50** | **50.37** | **50.37** | **55.15** | **49.17** | **49.04** | **49.15** | **47.46** | **47.46** | **47.38** | **47.79** |

### MuJoCo (MP: MSE)

| Method | $r_a = 0$ | | | | $r_a = 0.2$ | | | | $r_a = 0.5$ | | | | $r_a = 0.8$ | | | |
|---|---|---|---|---|---|---|---|---|---|---|---|---|---|---|---|---|
| | $r_b=0$ | $r_b=0.2$ | $r_b=0.5$ | $r_b=0.8$ | $r_b=0$ | $r_b=0.2$ | $r_b=0.5$ | $r_b=0.8$ | $r_b=0$ | $r_b=0.2$ | $r_b=0.5$ | $r_b=0.8$ | $r_b=0$ | $r_b=0.2$ | $r_b=0.5$ | $r_b=0.8$ |
| Base | 0.016 | 4.998 | 7.779 | 8.517 | 0.021 | 4.984 | 7.783 | 8.536 | 0.063 | 4.997 | 7.785 | 8.536 | 0.221 | 5.056 | 7.786 | 8.543 |
| LEEF-VFL [47] | 0.013 | 4.322 | 7.741 | 8.322 | 0.016 | 4.713 | 7.743 | 8.337 | 0.015 | 4.731 | 7.754 | 8.349 | 0.114 | 4.841 | 7.756 | 8.423 |
| LASER-VFL [53] | **0.012** | **0.103** | **0.103** | **0.104** | **0.014** | **0.108** | **0.109** | **0.110** | **0.014** | **0.108** | **0.108** | **0.109** | **0.108** | **0.129** | **0.130** | **0.131** |

### NUS-WIDE (MP: $\mathcal{A}$ (%))

| Method | $r_a = 0$ | | | | $r_a = 0.2$ | | | | $r_a = 0.5$ | | | | $r_a = 0.8$ | | | |
|---|---|---|---|---|---|---|---|---|---|---|---|---|---|---|---|---|
| | $r_b=0$ | $r_b=0.2$ | $r_b=0.5$ | $r_b=0.8$ | $r_b=0$ | $r_b=0.2$ | $r_b=0.5$ | $r_b=0.8$ | $r_b=0$ | $r_b=0.2$ | $r_b=0.5$ | $r_b=0.8$ | $r_b=0$ | $r_b=0.2$ | $r_b=0.5$ | $r_b=0.8$ |
| Base | **81.64** | 52.76 | 32.23 | 22.95 | 72.12 | 53.09 | 32.73 | 22.98 | 73.19 | 54.29 | 33.41 | 23.17 | 72.54 | 51.05 | 31.08 | 23.02 |
| LEEF-VFL [47] | 81.31 | **53.28** | **33.62** | **23.94** | **75.41** | **53.39** | **33.62** | **23.74** | **76.18** | **55.45** | **34.22** | **23.56** | **75.76** | **54.65** | **33.23** | **23.80** |

### CMU-MOSI (MP: $\mathcal{A}$ (%))

| Method | $r_a = 0$ | | | | $r_a = 0.2$ | | | | $r_a = 0.5$ | | | | $r_a = 0.8$ | | | |
|---|---|---|---|---|---|---|---|---|---|---|---|---|---|---|---|---|
| | $r_b=0$ | $r_b=0.2$ | $r_b=0.5$ | $r_b=0.8$ | $r_b=0$ | $r_b=0.2$ | $r_b=0.5$ | $r_b=0.8$ | $r_b=0$ | $r_b=0.2$ | $r_b=0.5$ | $r_b=0.8$ | $r_b=0$ | $r_b=0.2$ | $r_b=0.5$ | $r_b=0.8$ |
| Base | **56.85** | **52.76** | 51.94 | 51.94 | **62.58** | **58.49** | 52.97 | **52.97** | 62.58 | 58.49 | 52.97 | **52.97** | 62.58 | 58.49 | 52.97 | **52.97** |
| LEEF-VFL [47] | 52.56 | 52.56 | **55.83** | **53.58** | 58.22 | 55.48 | **54.79** | 48.88 | **62.58** | **59.30** | **58.08** | 48.88 | **62.58** | **59.30** | **58.08** | 48.88 |

### CMU-MOSEI (MP: $\mathcal{A}$ (%))

| Method | $r_a = 0$ | | | | $r_a = 0.2$ | | | | $r_a = 0.5$ | | | | $r_a = 0.8$ | | | |
|---|---|---|---|---|---|---|---|---|---|---|---|---|---|---|---|---|
| | $r_b=0$ | $r_b=0.2$ | $r_b=0.5$ | $r_b=0.8$ | $r_b=0$ | $r_b=0.2$ | $r_b=0.5$ | $r_b=0.8$ | $r_b=0$ | $r_b=0.2$ | $r_b=0.5$ | $r_b=0.8$ | $r_b=0$ | $r_b=0.2$ | $r_b=0.5$ | $r_b=0.8$ |
| Base | 60.89 | 54.31 | 49.19 | 49.34 | 57.93 | 49.83 | 47.47 | 47.92 | 52.53 | 47.92 | 46.66 | 47.16 | 47.79 | 46.44 | 46.30 | 44.78 |
| LEEF-VFL [47] | **60.95** | **55.20** | **49.59** | **49.80** | **58.19** | **51.22** | **47.62** | **48.20** | **53.02** | **48.16** | **46.78** | **47.97** | **48.14** | **47.86** | **47.72** | **47.32** |

### UR-FUNNY (MP: $\mathcal{A}$ (%))

| Method | $r_a = 0$ | | | | $r_a = 0.2$ | | | | $r_a = 0.5$ | | | | $r_a = 0.8$ | | | |
|---|---|---|---|---|---|---|---|---|---|---|---|---|---|---|---|---|
| | $r_b=0$ | $r_b=0.2$ | $r_b=0.5$ | $r_b=0.8$ | $r_b=0$ | $r_b=0.2$ | $r_b=0.5$ | $r_b=0.8$ | $r_b=0$ | $r_b=0.2$ | $r_b=0.5$ | $r_b=0.8$ | $r_b=0$ | $r_b=0.2$ | $r_b=0.5$ | $r_b=0.8$ |
| Base | 63.33 | 62.19 | 56.19 | 53.21 | 62.19 | 60.11 | 54.16 | 52.08 | 55.51 | 54.93 | 53.12 | 51.32 | 52.74 | 53.31 | 51.70 | 49.43 |
| LEEF-VFL [47] | **63.52** | **62.29** | **56.24** | **56.24** | **62.85** | **60.40** | **55.47** | **54.16** | **62.67** | **55.71** | **54.06** | **52.17** | **54.16** | **54.16** | **52.08** | **51.32** |

### MUSTARD (MP: $\mathcal{A}$ (%))

| Method | $r_a = 0$ | | | | $r_a = 0.2$ | | | | $r_a = 0.5$ | | | | $r_a = 0.8$ | | | |
|---|---|---|---|---|---|---|---|---|---|---|---|---|---|---|---|---|
| | $r_b=0$ | $r_b=0.2$ | $r_b=0.5$ | $r_b=0.8$ | $r_b=0$ | $r_b=0.2$ | $r_b=0.5$ | $r_b=0.8$ | $r_b=0$ | $r_b=0.2$ | $r_b=0.5$ | $r_b=0.8$ | $r_b=0$ | $r_b=0.2$ | $r_b=0.5$ | $r_b=0.8$ |
| Base | 57.97 | 54.34 | 47.83 | 48.55 | 56.52 | 50.72 | 44.20 | 43.48 | 55.07 | 47.83 | 42.75 | 42.75 | 50.00 | 46.37 | 42.75 | 39.86 |
| LEEF-VFL [47] | **58.70** | **55.80** | **51.45** | **49.28** | **57.25** | **52.17** | **44.93** | **44.93** | **55.79** | **49.28** | **43.48** | **43.48** | **52.17** | **48.55** | **44.93** | **43.48** |

### PTB-XL (MP: $F_1$ (%))

| Method | $r_a = 0$ | | | | $r_a = 0.2$ | | | | $r_a = 0.5$ | | | | $r_a = 0.8$ | | | |
|---|---|---|---|---|---|---|---|---|---|---|---|---|---|---|---|---|
| | $r_b=0$ | $r_b=0.2$ | $r_b=0.5$ | $r_b=0.8$ | $r_b=0$ | $r_b=0.2$ | $r_b=0.5$ | $r_b=0.8$ | $r_b=0$ | $r_b=0.2$ | $r_b=0.5$ | $r_b=0.8$ | $r_b=0$ | $r_b=0.2$ | $r_b=0.5$ | $r_b=0.8$ |
| Base | 57.25 | 33.25 | 33.83 | 33.73 | 53.50 | 33.55 | 33.47 | 33.17 | 48.40 | 32.50 | 33.05 | **33.36** | 30.03 | 28.54 | 32.36 | 32.22 |
| LEEF-VFL [47] | **57.60** | **37.62** | **36.39** | **34.08** | **55.00** | **35.90** | **35.67** | **34.08** | **49.31** | **34.02** | **34.42** | 33.33 | **36.53** | **34.02** | **34.42** | **32.88** |

### MIMIC-III (MP: $\mathcal{A}$ (%))

| Method | $r_a = 0$ | | | | $r_a = 0.2$ | | | | $r_a = 0.5$ | | | | $r_a = 0.8$ | | | |
|---|---|---|---|---|---|---|---|---|---|---|---|---|---|---|---|---|
| | $r_b=0$ | $r_b=0.2$ | $r_b=0.5$ | $r_b=0.8$ | $r_b=0$ | $r_b=0.2$ | $r_b=0.5$ | $r_b=0.8$ | $r_b=0$ | $r_b=0.2$ | $r_b=0.5$ | $r_b=0.8$ | $r_b=0$ | $r_b=0.2$ | $r_b=0.5$ | $r_b=0.8$ |
| Base | 61.10 | 54.80 | 52.66 | **50.51** | 60.70 | 58.09 | 53.31 | 49.36 | **59.40** | 57.16 | 53.77 | **50.57** | 56.32 | 55.45 | 51.76 | **50.36** |
| LEEF-VFL [47] | **61.54** | **58.68** | **55.48** | 49.95 | **61.17** | **58.25** | **57.53** | **50.57** | 59.06 | **58.59** | **56.91** | 49.64 | **57.66** | **56.57** | **52.13** | 49.43 |

### VISION&TOUCH (MP: $\mathcal{A}$ (%))

| Method | $r_a = 0$ | | | | $r_a = 0.2$ | | | | $r_a = 0.5$ | | | | $r_a = 0.8$ | | | |
|---|---|---|---|---|---|---|---|---|---|---|---|---|---|---|---|---|
| | $r_b=0$ | $r_b=0.2$ | $r_b=0.5$ | $r_b=0.8$ | $r_b=0$ | $r_b=0.2$ | $r_b=0.5$ | $r_b=0.8$ | $r_b=0$ | $r_b=0.2$ | $r_b=0.5$ | $r_b=0.8$ | $r_b=0$ | $r_b=0.2$ | $r_b=0.5$ | $r_b=0.8$ |
| Base | 92.39 | 90.15 | 89.57 | 86.96 | 91.15 | 88.09 | 86.05 | 83.14 | 90.25 | 85.03 | 82.81 | 78.29 | 88.79 | 83.27 | 80.84 | 75.38 |
| LEEF-VFL [47] | **92.51** | **90.59** | **90.08** | **88.38** | **91.88** | **88.49** | **86.68** | **84.49** | **90.52** | **86.73** | **83.32** | **79.22** | **89.06** | **84.98** | **81.52** | **78.21** |

### Table 19: **Results with Corrupted Features.**

**UCI-HAR (MP: $\mathcal{A}$ (%))**

| Method | $r_a=0$ | | | | $r_a=0.2$ | | | | $r_a=0.5$ | | | | $r_a=0.8$ | | | |
|---|---|---|---|---|---|---|---|---|---|---|---|---|---|---|---|---|
| | $r_b=0$ | $r_b=0.2$ | $r_b=0.5$ | $r_b=0.8$ | $r_b=0$ | $r_b=0.2$ | $r_b=0.5$ | $r_b=0.8$ | $r_b=0$ | $r_b=0.2$ | $r_b=0.5$ | $r_b=0.8$ | $r_b=0$ | $r_b=0.2$ | $r_b=0.5$ | $r_b=0.8$ |
| Base | **94.67** | 91.82 | 88.60 | 86.16 | 93.89 | 91.82 | 88.84 | 85.85 | **94.27** | 91.92 | 87.61 | 84.12 | 93.86 | 91.25 | 87.95 | 84.15 |
| RVFL-Aug | 93.99 | **93.21** | **90.23** | **88.19** | **94.13** | **92.20** | **90.53** | **87.89** | 94.13 | **92.23** | **88.63** | **85.99** | **94.91** | **91.48** | **88.29** | **85.37** |

**KU-HAR (MP: $\mathcal{A}$ (%))**

| Method | $r_a=0$ | | | | $r_a=0.2$ | | | | $r_a=0.5$ | | | | $r_a=0.8$ | | | |
|---|---|---|---|---|---|---|---|---|---|---|---|---|---|---|---|---|
| | $r_b=0$ | $r_b=0.2$ | $r_b=0.5$ | $r_b=0.8$ | $r_b=0$ | $r_b=0.2$ | $r_b=0.5$ | $r_b=0.8$ | $r_b=0$ | $r_b=0.2$ | $r_b=0.5$ | $r_b=0.8$ | $r_b=0$ | $r_b=0.2$ | $r_b=0.5$ | $r_b=0.8$ |
| Base | 82.24 | 76.58 | 69.98 | 63.76 | 80.58 | 77.04 | 71.40 | 65.40 | 79.37 | 75.83 | 71.37 | 66.72 | 76.58 | 74.10 | 69.73 | 66.12 |
| RVFL-Aug | **82.31** | **77.71** | **70.77** | **64.60** | **81.08** | **77.49** | **71.64** | **65.73** | **80.12** | **76.75** | **72.46** | **67.52** | **77.04** | **74.63** | **70.53** | **66.67** |

**MuJoCo (MP: MSE)**

| Method | $r_a=0$ | | | | $r_a=0.2$ | | | | $r_a=0.5$ | | | | $r_a=0.8$ | | | |
|---|---|---|---|---|---|---|---|---|---|---|---|---|---|---|---|---|
| | $r_b=0$ | $r_b=0.2$ | $r_b=0.5$ | $r_b=0.8$ | $r_b=0$ | $r_b=0.2$ | $r_b=0.5$ | $r_b=0.8$ | $r_b=0$ | $r_b=0.2$ | $r_b=0.5$ | $r_b=0.8$ | $r_b=0$ | $r_b=0.2$ | $r_b=0.5$ | $r_b=0.8$ |
| Base | 0.016 | 0.058 | 0.126 | 0.212 | 0.013 | 0.021 | 0.031 | 0.041 | **0.014** | 0.022 | 0.032 | 0.040 | 0.019 | 0.025 | 0.033 | 0.043 |
| RVFL-Aug | **0.013** | **0.051** | **0.110** | **0.196** | **0.013** | **0.019** | **0.030** | **0.039** | 0.015 | **0.022** | **0.028** | **0.040** | **0.017** | **0.021** | **0.031** | **0.041** |

**VISION&TOUCH (MP: $\mathcal{A}$ (%))**

| Method | $r_a=0$ | | | | $r_a=0.2$ | | | | $r_a=0.5$ | | | | $r_a=0.8$ | | | |
|---|---|---|---|---|---|---|---|---|---|---|---|---|---|---|---|---|
| | $r_b=0$ | $r_b=0.2$ | $r_b=0.5$ | $r_b=0.8$ | $r_b=0$ | $r_b=0.2$ | $r_b=0.5$ | $r_b=0.8$ | $r_b=0$ | $r_b=0.2$ | $r_b=0.5$ | $r_b=0.8$ | $r_b=0$ | $r_b=0.2$ | $r_b=0.5$ | $r_b=0.8$ |
| Base | 92.39 | 91.56 | 90.76 | 88.64 | 91.63 | 90.74 | 89.83 | 84.72 | 90.83 | 89.42 | 87.44 | 81.36 | 89.96 | 87.74 | 84.28 | 78.84 |
| RVFL-Aug | **92.58** | **92.23** | **91.58** | **89.73** | **92.06** | **91.59** | **90.26** | **87.36** | **91.78** | **91.03** | **88.76** | **84.06** | **91.24** | **90.22** | **87.25** | **81.42** |

**MIMIC-III (MP: $\mathcal{A}$ (%))**

| Method | $r_a=0$ | | | | $r_a=0.2$ | | | | $r_a=0.5$ | | | | $r_a=0.8$ | | | |
|---|---|---|---|---|---|---|---|---|---|---|---|---|---|---|---|---|
| | $r_b=0$ | $r_b=0.2$ | $r_b=0.5$ | $r_b=0.8$ | $r_b=0$ | $r_b=0.2$ | $r_b=0.5$ | $r_b=0.8$ | $r_b=0$ | $r_b=0.2$ | $r_b=0.5$ | $r_b=0.8$ | $r_b=0$ | $r_b=0.2$ | $r_b=0.5$ | $r_b=0.8$ |
| Base | 61.10 | 60.33 | 58.19 | 57.04 | 60.02 | 59.12 | 57.13 | 55.23 | 59.27 | 56.19 | 54.21 | 51.82 | 57.66 | 55.17 | 52.16 | 50.36 |
| RVFL-Aug | **61.85** | **61.51** | **59.12** | **58.09** | **61.45** | **60.73** | **57.44** | **55.79** | **59.83** | **57.01** | **55.79** | **53.59** | **58.06** | **56.32** | **54.24** | **52.24** |

**PTB-XL (MP: $F_1$ (%))**

| Method | $r_a=0$ | | | | $r_a=0.2$ | | | | $r_a=0.5$ | | | | $r_a=0.8$ | | | |
|---|---|---|---|---|---|---|---|---|---|---|---|---|---|---|---|---|
| | $r_b=0$ | $r_b=0.2$ | $r_b=0.5$ | $r_b=0.8$ | $r_b=0$ | $r_b=0.2$ | $r_b=0.5$ | $r_b=0.8$ | $r_b=0$ | $r_b=0.2$ | $r_b=0.5$ | $r_b=0.8$ | $r_b=0$ | $r_b=0.2$ | $r_b=0.5$ | $r_b=0.8$ |
| Base | 57.25 | 55.38 | 54.44 | 50.81 | 55.92 | 54.70 | 54.26 | 50.64 | 55.71 | 53.67 | 53.25 | 49.31 | 52.08 | 52.40 | 50.56 | 48.40 |
| RVFL-Aug | **58.45** | **55.92** | **55.73** | **51.20** | **56.61** | **55.05** | **55.27** | **50.77** | **56.13** | **54.75** | **53.60** | **50.09** | **54.43** | **53.26** | **50.61** | **49.24** |

**UR-FUNNY (MP: $\mathcal{A}$ (%))**

| Method | $r_a=0$ | | | | $r_a=0.2$ | | | | $r_a=0.5$ | | | | $r_a=0.8$ | | | |
|---|---|---|---|---|---|---|---|---|---|---|---|---|---|---|---|---|
| | $r_b=0$ | $r_b=0.2$ | $r_b=0.5$ | $r_b=0.8$ | $r_b=0$ | $r_b=0.2$ | $r_b=0.5$ | $r_b=0.8$ | $r_b=0$ | $r_b=0.2$ | $r_b=0.5$ | $r_b=0.8$ | $r_b=0$ | $r_b=0.2$ | $r_b=0.5$ | $r_b=0.8$ |
| Base | 63.33 | 63.04 | 62.85 | 62.29 | 62.76 | 62.95 | 62.38 | 62.19 | 62.38 | 62.67 | 62.48 | 61.53 | 61.63 | 62.85 | 62.29 | 61.44 |
| RVFL-Aug | **63.52** | **63.23** | **63.71** | **63.04** | **63.23** | **63.14** | **63.04** | **63.42** | **63.42** | **63.23** | **63.23** | **62.10** | **62.95** | **63.52** | **62.38** | **61.72** |

**MUSTARD (MP: $\mathcal{A}$ (%))**

| Method | $r_a=0$ | | | | $r_a=0.2$ | | | | $r_a=0.5$ | | | | $r_a=0.8$ | | | |
|---|---|---|---|---|---|---|---|---|---|---|---|---|---|---|---|---|
| | $r_b=0$ | $r_b=0.2$ | $r_b=0.5$ | $r_b=0.8$ | $r_b=0$ | $r_b=0.2$ | $r_b=0.5$ | $r_b=0.8$ | $r_b=0$ | $r_b=0.2$ | $r_b=0.5$ | $r_b=0.8$ | $r_b=0$ | $r_b=0.2$ | $r_b=0.5$ | $r_b=0.8$ |
| Base | 57.97 | 55.80 | **56.52** | 51.45 | 55.80 | 54.35 | 55.07 | 51.45 | 57.25 | 53.62 | 52.89 | 48.55 | 56.52 | 53.62 | 52.89 | 47.83 |
| RVFL-Aug | **58.70** | **56.52** | 55.80 | **52.90** | **56.52** | **55.07** | **55.80** | **52.17** | **57.97** | **56.52** | **54.35** | **50.72** | **58.70** | **55.07** | **53.62** | **49.27** |

**CMU-MOSI (MP: $\mathcal{A}$ (%))**

| Method | $r_a=0$ | | | | $r_a=0.2$ | | | | $r_a=0.5$ | | | | $r_a=0.8$ | | | |
|---|---|---|---|---|---|---|---|---|---|---|---|---|---|---|---|---|
| | $r_b=0$ | $r_b=0.2$ | $r_b=0.5$ | $r_b=0.8$ | $r_b=0$ | $r_b=0.2$ | $r_b=0.5$ | $r_b=0.8$ | $r_b=0$ | $r_b=0.2$ | $r_b=0.5$ | $r_b=0.8$ | $r_b=0$ | $r_b=0.2$ | $r_b=0.5$ | $r_b=0.8$ |
| Base | 56.85 | 55.97 | 54.22 | 47.37 | 54.37 | 54.08 | 52.47 | 46.64 | 52.76 | 53.49 | 49.85 | 43.97 | 53.06 | 51.53 | 46.06 | 37.63 |
| RVFL-Aug | **57.24** | **56.70** | **55.24** | **49.27** | **55.10** | **54.51** | **52.91** | **48.97** | **54.19** | **53.93** | **50.43** | **45.91** | **53.64** | **52.91** | **46.50** | **41.92** |

**CMU-MOSEI (MP: $\mathcal{A}$ (%))**

| Method | $r_a=0$ | | | | $r_a=0.2$ | | | | $r_a=0.5$ | | | | $r_a=0.8$ | | | |
|---|---|---|---|---|---|---|---|---|---|---|---|---|---|---|---|---|
| | $r_b=0$ | $r_b=0.2$ | $r_b=0.5$ | $r_b=0.8$ | $r_b=0$ | $r_b=0.2$ | $r_b=0.5$ | $r_b=0.8$ | $r_b=0$ | $r_b=0.2$ | $r_b=0.5$ | $r_b=0.8$ | $r_b=0$ | $r_b=0.2$ | $r_b=0.5$ | $r_b=0.8$ |
| Base | 60.89 | 56.76 | 57.72 | 56.71 | 60.54 | 52.29 | 56.84 | 55.83 | **60.68** | 50.89 | 50.76 | 51.26 | 55.59 | 48.34 | 47.42 | 48.43 |
| RVFL-Aug | **63.68** | **60.78** | **60.31** | **57.89** | **62.39** | **56.94** | **58.20** | **57.94** | 60.64 | **53.24** | **57.92** | **56.64** | **55.78** | **52.29** | **55.29** | **53.80** |

**NUS-WIDE (MP: $\mathcal{A}$ (%))**

| Method | $r_a=0$ | | | | $r_a=0.2$ | | | | $r_a=0.5$ | | | | $r_a=0.8$ | | | |
|---|---|---|---|---|---|---|---|---|---|---|---|---|---|---|---|---|
| | $r_b=0$ | $r_b=0.2$ | $r_b=0.5$ | $r_b=0.8$ | $r_b=0$ | $r_b=0.2$ | $r_b=0.5$ | $r_b=0.8$ | $r_b=0$ | $r_b=0.2$ | $r_b=0.5$ | $r_b=0.8$ | $r_b=0$ | $r_b=0.2$ | $r_b=0.5$ | $r_b=0.8$ |
| Base | 81.64 | 74.42 | 69.21 | 64.31 | 81.26 | 76.40 | 69.47 | 63.15 | 80.48 | 76.14 | 69.44 | 62.69 | 79.03 | 74.94 | 66.61 | 58.49 |
| RVFL-Aug | **82.17** | **75.71** | **70.44** | **64.54** | **81.92** | **77.32** | **70.37** | **64.04** | **81.13** | **76.81** | **70.43** | **63.55** | **80.47** | **76.29** | **67.34** | **59.24** |

**MM-IMDB (MP: $F_1$ (%))**

| Method | $r_a=0$ | | | | $r_a=0.2$ | | | | $r_a=0.5$ | | | | $r_a=0.8$ | | | |
|---|---|---|---|---|---|---|---|---|---|---|---|---|---|---|---|---|
| | $r_b=0$ | $r_b=0.2$ | $r_b=0.5$ | $r_b=0.8$ | $r_b=0$ | $r_b=0.2$ | $r_b=0.5$ | $r_b=0.8$ | $r_b=0$ | $r_b=0.2$ | $r_b=0.5$ | $r_b=0.8$ | $r_b=0$ | $r_b=0.2$ | $r_b=0.5$ | $r_b=0.8$ |
| Base | **56.35** | 45.85 | 34.86 | 33.48 | 39.51 | 35.90 | 34.67 | 32.16 | 39.46 | 37.24 | 31.31 | 27.95 | 37.47 | 36.13 | 25.49 | 15.26 |
| RVFL-Aug | 55.68 | **46.32** | **42.59** | **42.21** | **44.55** | **42.99** | **42.56** | **41.45** | **44.22** | **43.25** | **41.26** | **39.63** | **43.27** | **42.90** | **34.46** | **24.72** |

Table 20: **Results with Misaligned Features.**

*UCI-HAR (MP: $\mathcal{A}$ (%))*

| Method | $r_a = 0$ | | | | $r_a = 0.5$ | | | | $r_a = 0.8$ | | | | $r_a = 1$ | | | |
|---|---|---|---|---|---|---|---|---|---|---|---|---|---|---|---|---|
| | $r_b=0$ | $r_b=0.5$ | $r_b=0.8$ | $r_b=1$ | $r_b=0$ | $r_b=0.5$ | $r_b=0.8$ | $r_b=1$ | $r_b=0$ | $r_b=0.5$ | $r_b=0.8$ | $r_b=1$ | $r_b=0$ | $r_b=0.5$ | $r_b=0.8$ | $r_b=1$ |
| Base | **94.67** | 90.63 | 90.87 | 89.01 | 90.97 | 90.91 | 90.77 | 88.53 | 90.87 | 90.29 | 87.75 | 86.56 | 90.60 | 90.80 | 85.23 | 84.86 |
| RVFL-Align | 94.47 | **91.11** | **90.91** | **91.01** | **91.04** | **90.94** | **91.01** | **90.97** | **91.08** | **90.84** | **91.08** | **91.04** | **90.80** | **91.01** | **91.11** | **91.08** |

*KU-HAR (MP: $\mathcal{A}$ (%))*

| Method | $r_a = 0$ | | | | $r_a = 0.5$ | | | | $r_a = 0.8$ | | | | $r_a = 1$ | | | |
|---|---|---|---|---|---|---|---|---|---|---|---|---|---|---|---|---|
| | $r_b=0$ | $r_b=0.5$ | $r_b=0.8$ | $r_b=1$ | $r_b=0$ | $r_b=0.5$ | $r_b=0.8$ | $r_b=1$ | $r_b=0$ | $r_b=0.5$ | $r_b=0.8$ | $r_b=1$ | $r_b=0$ | $r_b=0.5$ | $r_b=0.8$ | $r_b=1$ |
| Base | **82.24** | 70.99 | 70.48 | 70.17 | **73.11** | 69.18 | 68.05 | 68.29 | 70.39 | 69.20 | 68.12 | 68.31 | 70.29 | 59.33 | 47.40 | 40.24 |
| RVFL-Align | 81.20 | **71.47** | **70.77** | **70.65** | 72.96 | **71.18** | **71.16** | **71.23** | **71.16** | **70.80** | **70.75** | **70.77** | **70.72** | **68.58** | **68.17** | **62.31** |

*MuJoCo (MP: MSE)*

| Method | $r_a = 0$ | | | | $r_a = 0.5$ | | | | $r_a = 0.8$ | | | | $r_a = 1$ | | | |
|---|---|---|---|---|---|---|---|---|---|---|---|---|---|---|---|---|
| | $r_b=0$ | $r_b=0.5$ | $r_b=0.8$ | $r_b=1$ | $r_b=0$ | $r_b=0.5$ | $r_b=0.8$ | $r_b=1$ | $r_b=0$ | $r_b=0.5$ | $r_b=0.8$ | $r_b=1$ | $r_b=0$ | $r_b=0.5$ | $r_b=0.8$ | $r_b=1$ |
| Base | **0.016** | 0.210 | 0.217 | 0.218 | 0.068 | 0.182 | 0.232 | 0.289 | 0.183 | 0.194 | 0.264 | 0.304 | 0.183 | 0.215 | 0.335 | 0.410 |
| RVFL-Align | 0.017 | **0.138** | **0.165** | **0.173** | **0.053** | **0.150** | **0.183** | **0.183** | **0.142** | **0.167** | **0.181** | **0.182** | **0.182** | **0.183** | **0.183** | **0.182** |

*VISION&TOUCH (MP: $\mathcal{A}$ (%))*

| Method | $r_a = 0$ | | | | $r_a = 0.5$ | | | | $r_a = 0.8$ | | | | $r_a = 1$ | | | |
|---|---|---|---|---|---|---|---|---|---|---|---|---|---|---|---|---|
| | $r_b=0$ | $r_b=0.5$ | $r_b=0.8$ | $r_b=1$ | $r_b=0$ | $r_b=0.5$ | $r_b=0.8$ | $r_b=1$ | $r_b=0$ | $r_b=0.5$ | $r_b=0.8$ | $r_b=1$ | $r_b=0$ | $r_b=0.5$ | $r_b=0.8$ | $r_b=1$ |
| Base | **92.39** | 85.36 | 81.68 | **79.48** | 84.62 | 79.64 | 78.46 | 77.64 | 76.38 | 73.28 | 74.34 | 72.24 | 70.42 | 70.86 | 69.83 | 66.38 |
| RVFL-Align | 91.83 | **86.73** | **82.39** | 79.32 | **86.56** | **80.13** | **79.23** | **77.86** | **77.49** | **74.64** | **75.18** | **73.58** | **71.08** | **71.13** | **70.05** | **67.12** |

*MIMIC-III (MP: $\mathcal{A}$ (%))*

| Method | $r_a = 0$ | | | | $r_a = 0.5$ | | | | $r_a = 0.8$ | | | | $r_a = 1$ | | | |
|---|---|---|---|---|---|---|---|---|---|---|---|---|---|---|---|---|
| | $r_b=0$ | $r_b=0.5$ | $r_b=0.8$ | $r_b=1$ | $r_b=0$ | $r_b=0.5$ | $r_b=0.8$ | $r_b=1$ | $r_b=0$ | $r_b=0.5$ | $r_b=0.8$ | $r_b=1$ | $r_b=0$ | $r_b=0.5$ | $r_b=0.8$ | $r_b=1$ |
| Base | **61.10** | 57.72 | 54.49 | 49.95 | 56.32 | 56.14 | 56.07 | 55.20 | 56.20 | 56.04 | 55.94 | 54.02 | 56.66 | 56.54 | 55.23 | 53.09 |
| RVFL-Align | 60.27 | **57.97** | **57.56** | **57.56** | **57.28** | **57.19** | **57.69** | **56.91** | **57.22** | **57.63** | **57.13** | **56.94** | **57.38** | **58.25** | **57.72** | **53.46** |

*PTB-XL (MP: $F_1$ (%))*

| Method | $r_a = 0$ | | | | $r_a = 0.5$ | | | | $r_a = 0.8$ | | | | $r_a = 1$ | | | |
|---|---|---|---|---|---|---|---|---|---|---|---|---|---|---|---|---|
| | $r_b=0$ | $r_b=0.5$ | $r_b=0.8$ | $r_b=1$ | $r_b=0$ | $r_b=0.5$ | $r_b=0.8$ | $r_b=1$ | $r_b=0$ | $r_b=0.5$ | $r_b=0.8$ | $r_b=1$ | $r_b=0$ | $r_b=0.5$ | $r_b=0.8$ | $r_b=1$ |
| Base | **57.25** | 31.79 | 31.26 | 31.09 | 30.82 | 30.65 | 30.97 | 30.87 | 31.38 | 31.20 | 31.46 | 31.35 | 31.34 | 30.69 | 31.10 | 30.65 |
| RVFL-Align | 53.50 | **36.71** | **31.45** | **31.46** | **43.59** | **39.78** | **36.28** | **34.11** | **33.41** | **33.24** | **33.13** | **33.66** | **32.37** | **32.03** | **31.37** | **31.35** |

*UR-FUNNY (MP: $\mathcal{A}$ (%))*

| Method | $r_a = 0$ | | | | $r_a = 0.5$ | | | | $r_a = 0.8$ | | | | $r_a = 1$ | | | |
|---|---|---|---|---|---|---|---|---|---|---|---|---|---|---|---|---|
| | $r_b=0$ | $r_b=0.5$ | $r_b=0.8$ | $r_b=1$ | $r_b=0$ | $r_b=0.5$ | $r_b=0.8$ | $r_b=1$ | $r_b=0$ | $r_b=0.5$ | $r_b=0.8$ | $r_b=1$ | $r_b=0$ | $r_b=0.5$ | $r_b=0.8$ | $r_b=1$ |
| Base | **63.33** | 55.77 | 54.06 | 53.21 | 58.60 | 57.18 | 56.90 | 55.39 | 58.79 | 58.41 | 57.94 | 56.43 | 58.79 | 58.70 | 57.37 | 58.13 |
| RVFL-Align | 62.77 | **58.79** | **58.41** | **56.62** | **60.49** | **58.13** | **57.84** | **58.13** | **60.68** | **58.51** | **58.79** | **57.94** | **58.88** | **59.74** | **57.94** | **58.51** |

*MUSTARD (MP: $\mathcal{A}$ (%))*

| Method | $r_a = 0$ | | | | $r_a = 0.5$ | | | | $r_a = 0.8$ | | | | $r_a = 1$ | | | |
|---|---|---|---|---|---|---|---|---|---|---|---|---|---|---|---|---|
| | $r_b=0$ | $r_b=0.5$ | $r_b=0.8$ | $r_b=1$ | $r_b=0$ | $r_b=0.5$ | $r_b=0.8$ | $r_b=1$ | $r_b=0$ | $r_b=0.5$ | $r_b=0.8$ | $r_b=1$ | $r_b=0$ | $r_b=0.5$ | $r_b=0.8$ | $r_b=1$ |
| Base | **57.97** | 55.07 | 55.07 | 53.62 | **56.52** | 54.35 | 53.62 | 52.90 | 54.35 | 54.35 | **54.35** | 52.90 | 52.17 | 52.17 | 51.45 | 50.72 |
| RVFL-Align | 56.52 | **56.52** | **55.80** | **55.80** | 55.80 | **55.80** | **54.35** | **54.35** | **57.25** | **55.07** | 53.62 | **53.62** | **55.80** | **53.62** | **52.90** | **52.90** |

*CMU-MOSI (MP: $\mathcal{A}$ (%))*

| Method | $r_a = 0$ | | | | $r_a = 0.5$ | | | | $r_a = 0.8$ | | | | $r_a = 1$ | | | |
|---|---|---|---|---|---|---|---|---|---|---|---|---|---|---|---|---|
| | $r_b=0$ | $r_b=0.5$ | $r_b=0.8$ | $r_b=1$ | $r_b=0$ | $r_b=0.5$ | $r_b=0.8$ | $r_b=1$ | $r_b=0$ | $r_b=0.5$ | $r_b=0.8$ | $r_b=1$ | $r_b=0$ | $r_b=0.5$ | $r_b=0.8$ | $r_b=1$ |
| Base | **56.85** | 49.08 | 49.08 | 48.67 | 49.49 | 47.44 | 47.24 | 46.63 | 46.42 | 46.22 | 45.60 | 45.60 | 46.22 | 45.81 | 43.56 | 42.74 |
| RVFL-Align | 56.24 | **53.37** | **53.37** | **53.58** | **52.76** | **51.46** | **48.47** | **48.88** | **50.44** | **50.44** | **47.96** | **46.65** | **50.10** | **48.06** | **45.48** | **44.61** |

*CMU-MOSEI (MP: $\mathcal{A}$ (%))*

| Method | $r_a = 0$ | | | | $r_a = 0.5$ | | | | $r_a = 0.8$ | | | | $r_a = 1$ | | | |
|---|---|---|---|---|---|---|---|---|---|---|---|---|---|---|---|---|
| | $r_b=0$ | $r_b=0.5$ | $r_b=0.8$ | $r_b=1$ | $r_b=0$ | $r_b=0.5$ | $r_b=0.8$ | $r_b=1$ | $r_b=0$ | $r_b=0.5$ | $r_b=0.8$ | $r_b=1$ | $r_b=0$ | $r_b=0.5$ | $r_b=0.8$ | $r_b=1$ |
| Base | **60.89** | 44.12 | 43.11 | 42.30 | 54.71 | 44.07 | 43.06 | 42.25 | 49.44 | 43.97 | 43.06 | 42.20 | 47.11 | 43.92 | 42.60 | 42.20 |
| RVFL-Align | 58.35 | **44.93** | **45.09** | **43.16** | **55.12** | **44.73** | **44.98** | **43.16** | **50.16** | **44.58** | **44.98** | **43.11** | **49.17** | **44.38** | **44.83** | **42.81** |

*NUS-WIDE (MP: $\mathcal{A}$ (%))*

| Method | $r_a = 0$ | | | | $r_a = 0.5$ | | | | $r_a = 0.8$ | | | | $r_a = 1$ | | | |
|---|---|---|---|---|---|---|---|---|---|---|---|---|---|---|---|---|
| | $r_b=0$ | $r_b=0.5$ | $r_b=0.8$ | $r_b=1$ | $r_b=0$ | $r_b=0.5$ | $r_b=0.8$ | $r_b=1$ | $r_b=0$ | $r_b=0.5$ | $r_b=0.8$ | $r_b=1$ | $r_b=0$ | $r_b=0.5$ | $r_b=0.8$ | $r_b=1$ |
| Base | **81.64** | 53.49 | **53.93** | 53.77 | 54.02 | 54.01 | 53.49 | 53.36 | **53.53** | 53.54 | 53.79 | 52.46 | 53.77 | 53.77 | 50.02 | 49.95 |
| RVFL-Align | 81.05 | **58.89** | 53.49 | **53.91** | **54.32** | **57.87** | **54.01** | **53.53** | 55.34 | **53.67** | **53.91** | **54.01** | **53.92** | **53.91** | **53.58** | **53.50** |

*MM-IMDB (MP: $F_1$ (%))*

| Method | $r_a = 0$ | | | | $r_a = 0.5$ | | | | $r_a = 0.8$ | | | | $r_a = 1$ | | | |
|---|---|---|---|---|---|---|---|---|---|---|---|---|---|---|---|---|
| | $r_b=0$ | $r_b=0.5$ | $r_b=0.8$ | $r_b=1$ | $r_b=0$ | $r_b=0.5$ | $r_b=0.8$ | $r_b=1$ | $r_b=0$ | $r_b=0.5$ | $r_b=0.8$ | $r_b=1$ | $r_b=0$ | $r_b=0.5$ | $r_b=0.8$ | $r_b=1$ |
| Base | **56.35** | 51.83 | 52.11 | 51.43 | 55.18 | 51.69 | 50.72 | 51.12 | 53.25 | 51.96 | 51.76 | 50.08 | 51.33 | 51.26 | 51.43 | 49.93 |
| RVFL-Align | 55.84 | **52.40** | **52.17** | **51.93** | **55.67** | **51.96** | **51.54** | **51.88** | **54.94** | **52.10** | **51.89** | **51.78** | **52.55** | **51.55** | **51.67** | **51.72** |

Table 21: **Results of Defenses on the UCI-HAR Dataset.** 'None' indicates attacks without defenses.

| | Noisy Gradients (NG) | | | | | | | | | | | |
| | Main Task Performance ($\mathcal{A}$ %) | | | | | | Attack Performance ( %) | | | | | |
| Noisy Scale | Label Inference | | Feature Inference | | Backdoor | | Label Inference | | Feature Inference | | Backdoor | |
| | PMC [15] | AMC [15] | GRNA [38] | MIA [30] | TECB [8] | LFBA [46] | PMC [15] | AMC [15] | GRNA [38] | MIA [30] | TECB [8] | LFBA [46] |
|---|---|---|---|---|---|---|---|---|---|---|---|---|
| None | **94.67** | 92.91 | **93.28** | 92.50 | 89.11 | **92.16** | 60.50 | **64.27** | 38.55 | **74.45** | **100** | 99.39 |
| 0.0001 | 91.55 | **91.89** | **91.48** | 91.45 | 89.82 | **91.24** | 55.34 | **60.50** | 38.05 | **74.45** | **100** | 89.05 |
| 0.001 | **90.02** | 88.02 | **89.45** | 88.32 | 77.60 | **79.26** | 55.38 | **56.02** | 37.00 | **74.40** | **97.70** | 81.30 |
| 0.01 | **73.09** | 72.51 | **73.60** | 72.04 | **74.69** | 73.26 | 51.75 | **52.73** | 33.40 | **74.15** | **16.26** | 28.14 |
| 0.1 | 34.31 | **46.01** | **38.21** | 37.19 | **44.96** | 17.88 | 54.59 | **55.78** | 28.45 | **73.75** | 17.24 | **20.52** |

| | Gradient Compression (GC) | | | | | | | | | | | |
| | Main Task Performance ($\mathcal{A}$ %) | | | | | | Attack Performance (%) | | | | | |
| Compression Rate | Label Inference | | Feature Inference | | Backdoor | | Label Inference | | Feature Inference | | Backdoor | |
| | PMC [15] | AMC [15] | GRNA [38] | MIA [30] | TECB [8] | LFBA [46] | PMC [15] | AMC [15] | GRNA [38] | MIA [30] | TECB [8] | LFBA [46] |
|---|---|---|---|---|---|---|---|---|---|---|---|---|
| None | **94.67** | 92.91 | **93.28** | 92.50 | 89.11 | **92.16** | 60.50 | **64.27** | 38.55 | **74.45** | **100** | 99.39 |
| 0.9 | **91.82** | 91.38 | **91.45** | 91.31 | 75.19 | **78.35** | 59.99 | **64.10** | 38.35 | **74.45** | **100** | 86.15 |
| 0.75 | **89.45** | 89.28 | 88.63 | **89.21** | **75.40** | 72.14 | **65.05** | 63.08 | 36.40 | **74.40** | **100** | 67.12 |
| 0.5 | 81.03 | **87.11** | **82.15** | 81.68 | **65.31** | 62.27 | 58.81 | **60.06** | 40.60 | **74.25** | **100** | 62.50 |
| 0.25 | 64.61 | **66.44** | 66.54 | **67.39** | 37.66 | **48.39** | 63.52 | 62.13 | 39.25 | **73.70** | 23.18 | **50.42** |

| | Privacy-perserving Deep Learning (PPDL) | | | | | | | | | | | |
| | Main Task Performance ($\mathcal{A}$ %) | | | | | | Attack Performance (%) | | | | | |
| Retention Rate | Label Inference | | Feature Inference | | Backdoor | | Label Inference | | Feature Inference | | Backdoor | |
| | PMC [15] | AMC [15] | GRNA [38] | MIA [30] | TECB [8] | LFBA [46] | PMC [15] | AMC [15] | GRNA [38] | MIA [30] | TECB [8] | LFBA [46] |
|---|---|---|---|---|---|---|---|---|---|---|---|---|
| None | **94.67** | 92.91 | **93.28** | 92.50 | 89.11 | **92.16** | 60.50 | 64.27 | 38.55 | **74.45** | **100** | 99.39 |
| 0.9 | 83.14 | **88.67** | **84.63** | 83.31 | 83.81 | **91.65** | **58.67** | 56.43 | 38.05 | **74.45** | **89.16** | 71.12 |
| 0.75 | 82.22 | **88.67** | **83.47** | 82.02 | 78.52 | **83.41** | **58.67** | 56.40 | 37.55 | **74.30** | **71.11** | 65.32 |
| 0.5 | 87.28 | **88.50** | **84.32** | 83.07 | **81.51** | 75.09 | 58.23 | **58.67** | 38.60 | **74.35** | **82.62** | 58.33 |
| 0.25 | 36.27 | **36.51** | **37.50** | 35.53 | **18.22** | 18.15 | 56.50 | **59.48** | 37.40 | **74.15** | 16.83 | **20.81** |

| | DiscreteSGD (DSGD) | | | | | | | | | | | |
| | Main Task Performance ($\mathcal{A}$ %) | | | | | | Attack Performance (%) | | | | | |
| Intervals | Label Inference | | Feature Inference | | Backdoor | | Label Inference | | Feature Inference | | Backdoor | |
| | PMC [15] | AMC [15] | GRNA [38] | MIA [30] | TECB [8] | LFBA [46] | PMC [15] | AMC [15] | GRNA [38] | MIA [30] | TECB [8] | LFBA [46] |
|---|---|---|---|---|---|---|---|---|---|---|---|---|
| None | **94.67** | 92.91 | **93.28** | 92.50 | 89.11 | **92.16** | 60.50 | **64.27** | 38.55 | **74.45** | **100** | 99.39 |
| 24 | 37.84 | **53.61** | **38.65** | 37.56 | 32.16 | **37.26** | 59.99 | **63.18** | 38.65 | **74.40** | 30.64 | **65.80** |
| 18 | **50.25** | 45.71 | 46.35 | **47.13** | 29.18 | **56.60** | 59.31 | **61.15** | 42.05 | **74.45** | 28.25 | **63.52** |
| 12 | **35.12** | 35.02 | 36.51 | **38.04** | 28.63 | **38.21** | 58.30 | 58.09 | 35.40 | **74.20** | 21.37 | **55.07** |
| 6 | 18.46 | **19.58** | 21.24 | **22.57** | 18.22 | **25.08** | **57.72** | 56.26 | 34.90 | **73.65** | 16.83 | **46.35** |

Table 22: **Results of Defenses on the NUS-WIDE Dataset.** 'None' indicates attacks without defenses.

| Noisy Scale | Noisy Gradients (NG) | | | | | | | | | | | |
|---|---|---|---|---|---|---|---|---|---|---|---|---|
| | Main Task Performance (𝒜 %) | | | | | | Attack Performance (%) | | | | | |
| | Label Inference | | Feature Inference | | Backdoor | | Label Inference | | Feature Inference | | Backdoor | |
| | PMC [15] | AMC [15] | GRNA [38] | MIA [30] | TECB [8] | LFBA [46] | PMC [15] | AMC [15] | GRNA [38] | MIA [30] | TECB [8] | LFBA [46] |
| None | **81.42** | 80.46 | **81.64** | 81.33 | 74.99 | **81.07** | 57.43 | **60.23** | 36.25 | **99.75** | **100** | 99.93 |
| 0.0001 | **80.59** | 79.05 | **81.16** | 80.62 | 62.18 | **78.35** | 52.81 | **54.45** | 28.90 | **95.80** | 31.20 | **86.23** |
| 0.01 | **77.64** | 77.56 | **79.56** | 78.91 | 45.55 | **76.45** | **56.43** | 52.14 | 39.60 | **95.95** | 26.40 | **73.08** |
| 0.01 | **66.81** | 62.88 | **65.52** | 63.28 | 42.83 | **64.17** | **45.29** | 45.00 | 37.20 | **95.10** | 23.64 | **55.69** |
| 0.1 | 54.22 | **58.16** | **58.71** | 58.23 | 37.66 | **54.16** | 41.78 | **43.26** | 34.40 | **94.70** | **21.58** | 18.13 |

| Compression Rate | Gradient Compression (GC) | | | | | | | | | | | |
|---|---|---|---|---|---|---|---|---|---|---|---|---|
| | Main Task Performance (𝒜 %) | | | | | | Attack Performance (%) | | | | | |
| | Label Inference | | Feature Inference | | Backdoor | | Label Inference | | Feature Inference | | Backdoor | |
| | PMC [15] | AMC [15] | GRNA [38] | MIA [30] | TECB [8] | LFBA [46] | PMC [15] | AMC [15] | GRNA [38] | MIA [30] | TECB [8] | LFBA [46] |
| None | **81.42** | 80.46 | **81.64** | 81.33 | 74.99 | **81.07** | 57.43 | **60.23** | 36.25 | **99.75** | **100** | 99.93 |
| 0.9 | **79.36** | 78.20 | **80.72** | 79.43 | 65.88 | **78.26** | 57.11 | **59.26** | 35.50 | **95.90** | **98.73** | 93.24 |
| 0.75 | **77.48** | 76.03 | **79.37** | 78.37 | 42.96 | **76.63** | 56.43 | **58.47** | 34.65 | **95.95** | **97.62** | 82.37 |
| 0.5 | 74.24 | **75.53** | **77.16** | 73.53 | 40.95 | **67.41** | **55.81** | 53.31 | 35.90 | **96.00** | **88.76** | 57.51 |
| 0.25 | 71.28 | **72.39** | **73.17** | 69.54 | 37.66 | **55.32** | 54.72 | **55.28** | 35.50 | **92.90** | 21.58 | **48.24** |

| Retention Rate | Privacy-perserving Deep Learning (PPDL) | | | | | | | | | | | |
|---|---|---|---|---|---|---|---|---|---|---|---|---|
| | Main Task Performance (𝒜 %) | | | | | | Attack Performance (%) | | | | | |
| | Label Inference | | Feature Inference | | Backdoor | | Label Inference | | Feature Inference | | Backdoor | |
| | PMC [15] | AMC [15] | GRNA [38] | MIA [30] | TECB [8] | LFBA [46] | PMC [15] | AMC [15] | GRNA [38] | MIA [30] | TECB [8] | LFBA [46] |
| None | **81.42** | 80.46 | **81.64** | 81.33 | 74.99 | **81.07** | 57.43 | **60.23** | 36.25 | **99.75** | **100** | 99.93 |
| 0.9 | 76.44 | **78.53** | 76.25 | **77.19** | 72.16 | **75.18** | **57.47** | 52.34 | 35.70 | **96.00** | 86.21 | **90.64** |
| 0.75 | **75.38** | 73.16 | **74.21** | 73.12 | 71.82 | **73.62** | **56.46** | 52.41 | 35.35 | **95.90** | **73.54** | 68.79 |
| 0.5 | **72.63** | 68.73 | 71.35 | **72.16** | 70.03 | **72.31** | **56.10** | 55.16 | 38.20 | **95.70** | **81.36** | 57.43 |
| 0.25 | 42.86 | **43.13** | 38.92 | **39.81** | 37.66 | **39.12** | 32.48 | **33.60** | 35.55 | **95.40** | 21.58 | **49.52** |

| Intervals | DiscreteSGD (DSGD) | | | | | | | | | | | |
|---|---|---|---|---|---|---|---|---|---|---|---|---|
| | Main Task Performance (𝒜 %) | | | | | | Attack Performance (%) | | | | | |
| | Label Inference | | Feature Inference | | Backdoor | | Label Inference | | Feature Inference | | Backdoor | |
| | PMC [15] | AMC [15] | GRNA [38] | MIA [30] | TECB [8] | LFBA [46] | PMC [15] | AMC [15] | GRNA [38] | MIA [30] | TECB [8] | LFBA [46] |
| None | **81.42** | 80.46 | **81.64** | 81.33 | 74.99 | **81.07** | 57.43 | **60.23** | 36.25 | **99.75** | **100** | 99.93 |
| 24 | 62.73 | **66.29** | **63.16** | 62.31 | 58.24 | **61.23** | 54.32 | **55.26** | 35.45 | **95.05** | 56.26 | **83.47** |
| 18 | 64.36 | **64.58** | **64.21** | 63.14 | 59.12 | **60.17** | 49.83 | **51.18** | 34.80 | **94.85** | 48.73 | **61.06** |
| 12 | **52.19** | 50.16 | **51.01** | 49.88 | 43.19 | **44.01** | 46.42 | **48.31** | 36.35 | **94.40** | 35.01 | **54.11** |
| 6 | 42.84 | **47.62** | **46.54** | 44.31 | 37.66 | **38.94** | 43.13 | **48.64** | 29.15 | **93.70** | 21.58 | **32.24** |

# F    Discussion

**Contribution.** While vertical federated learning (VFL) has emerged as a critical privacy-preserving paradigm and achieved significant progress [18, 37, 69, 42, 23, 56], MARS-VFL offers a unified and comprehensive benchmark for systematically evaluating VFL methods. It includes a variety of datasets and implements realistic evaluation protocols that align with VFL settings. Although many VFL methods have publicly released their code, comparing them remains challenging due to inconsistencies in experimental settings, datasets, and implementation details. MARS-VFL addresses these issues by providing standardized implementations and evaluation settings across different methods, thereby improving reproducibility and fair comparison. With its modular design and broad coverage, MARS-VFL also facilitates the easy extension and integration of new methods, making it a valuable resource for both researchers and practitioners.

**Limitation and Future Work.** Despite the strengths of MARS-VFL, several limitations remain, which also open up promising directions for future works:

*(1) More datasets, models, and evaluation metrics.* Although MARS-VFL includes diverse datasets and representative models, it may not cover the full spectrum of real-world VFL applications. Some important datasets, model architectures, or task-specific metrics may be missing. However, MARS-VFL is designed to be extensible, and we plan to continuously update it by integrating more components in future releases.

*(2) Coverage of baseline methods.* Given the rapid development of VFL research, it is difficult to include every recent method. Some newly proposed approaches may not yet be incorporated into the benchmark. In this release, we focus on representative and publicly available methods. We welcome community contributions and will continue to expand the benchmark with newly published and open-source baselines.

*(3) Other research problems in VFL.* While MARS-VFL incorporates evaluation aspects related to efficiency, robustness, and security, covering a broad range of existing VFL methods, several significant research problems remain outside the scope of the current version: (i) Fairness in collaborative learning: Fairness concerns arise when clients contribute unequally to the collaboration due to data imbalance or heterogeneous feature distributions [23, 5, 29, 14]. Although fairness-aware learning has gained attention with unified evaluations in horizontal FL [32, 9, 22], it lacks standardized evaluation protocols in VFL settings. (ii) Asynchronous training: Most existing benchmarks assume synchronous collaboration, whereas real-world VFL systems often require asynchronous training to accommodate communication delays, stragglers, and different computational capabilities across clients [74, 73, 48, 72]. While MARS-VFL provides a robust and extensible pipeline for benchmarking VFL methods, incorporating these additional aspects would offer a more comprehensive and realistic foundation for advancing VFL research.

