# OpenReview forum: "MARS-VFL: A Unified Benchmark for Vertical Federated Learning with Realistic Evaluation"
_NeurIPS.cc/2025/Datasets_and_Benchmarks_Track — NeurIPS 2025 Datasets and Benchmarks Track spotlight_

### Official Review · Reviewer_wZVx · 2025-06-03

**Rating:** 6
**Confidence:** 4

**Summary:**

The paper constructs a new benchmark for vertical federated learning (VFL) that uses practical, real-world data. It includes a variety of datasets from real-world applications, which avoid the need to artificially divide raw data. Evaluations of several methods are carried out and it firstly assesses robustness in the VFL context. The authors also propose a new method to handle challenges of feature missing and feature misalignment.

**Additional Feedback:**

Please see the weaknesses.

**Dataset Code Accessibility:**

Yes

**Dataset Code Comments:**

The data and executable code are provided.

**Ethical Considerations:**

No, there are no or only very minor ethics concerns

**Final Justification:**

My quetions have all been addressed.

**Limitations Weaknesses:**

My concerns mainly concentrate on the experiments and the selection of datasets:

W1: In Figure 5, we would generally expect the attack success rate to decrease as defense strength increases. However, some methods (e.g., MIA) maintain a stable attack success rate. Could the authors clarify why these methods appear more resistant to defenses?

W2: Although RVFL shows better performance against different types of perturbations (as shown in Tables 4 and 5), its performance on clean data is lower than that of the base method. Does this indicate a trade-off in which robustness comes at the cost of general performance under normal conditions?

W3: Are the attack methods evaluated on datasets beyond the two mentioned datasets? Including experiments on other datasets would help assess how well these methods generalize to other cases.

W4: It would be helpful to provide more explanations about the selection criteria for these datasets. The authors just explain the suitability of the selected datasets, but not discuss the criteria to select them. Without a clear illustration, it is difficult to assess how representative they are in VFL applications.

W5: Minor issues. In Figure 2 and 3, there should be periods in the captions. In Table 19 and Table 20, missing spaces in table header of dataset name.

**Strengths Contributions:**

The main contributions of this paper are the discussion of robustness challenges and the use of practical data:

S1: I appreciate that this work is the first to discuss robustness problems in VFL. It broadens the scope of research fields and helps to uncover new research directions. The proposed method is backed by rigorous experiments, showing its effectiveness.

S2: It evaluates VFL methods using data from real-world sources. Unlike setups based on artificially partitioned data, this natural partition allows for a more intuitive evaluation of different methods. Besides, the benchmark supports different data types for flexibility and practical use.

S3: It evaluates methods using unified metrics. It is important for enabling comparisons between different methods. The figures and tables are presented clearly and concisely. Overall, the paper is well-organized and easy to understand.

---

> ### Author Rebuttal · Authors · 2025-07-30
>
> Thank you for your support and constructive comments. We have provided detailed responses below and would be happy to engage in further discussion regarding any of these points.
>
> > In Figure 5, we would generally expect the attack success rate to decrease as defense strength increases. However, some methods (e.g., MIA) maintain a stable attack success rate. Could the authors clarify why these methods appear more resistant to defenses?
>
> Response: Thank you for your valuable comments. We would like to clarify that while our evaluation of gradient-based defenses follows the experimental settings of prior work [1], certain attack methods based on embedding reconstruction (such as MIA) may remain effective, which are free from the infulence of gradients. We will incorporate additional defense methods in future versions to enable more comprehensive security evaluations.
>
> > Although RVFL shows better performance against different types of perturbations (as shown in Tables 4 and 5), its performance on clean data is lower than that of the base method. Does this indicate a trade-off in which robustness comes at the cost of general performance under normal conditions?
>
> Response: Thank you for raising this important issue. While RVFL demonstrates strong generalization against various perturbations, its performance on clean data may be degraded due to: (1) information loss from data augmentation, or (2) accidental shuffling of well-aligned samples. These factors can lead to performance drops and warrant deeper investigation through ablation studies in future work. In the current version, we establish a simple robustness baseline that can be readily extended for more comprehensive analyses.
>
> > Are the attack methods evaluated on datasets beyond the two mentioned datasets? Including experiments on other datasets would help assess how well these methods generalize to other cases.
>
> Response: Thank you for your constructive comments. We have developed basic evaluation pipelines using two representative datasets for security assessment, which can be readily extended to other datasets. As complementary evaluations, we provide additional results on the UR-FUNNY dataset that further support the conclusions presented in our paper:
>
> |    |  |                   |                   |                   |                   |                   |
> |----------|---------------------------|-------------------|-------------------|-------------------|-------------------|-------------------|
> |    **Metric**      | *PMC*       | *AMC*       | *GRNA*      | *MIA*       | *TECB*      | *LFBA*      |
> | **MP**   | 63.33     | 59.45     | 62.19     | 60.11     | 58.22     | 61.72     |
> | **AP**   | 42.16     | 44.52     | 34.40     | 71.46     | 100.00    | 98.26     |
>
> > It would be helpful to provide more explanations about the selection criteria for these datasets. The authors just explain the suitability of the selected datasets, but not discuss the criteria to select them. Without a clear illustration, it is difficult to assess how representative they are in VFL applications.
>
> Response: Thank you for your insightful comments. To construct evaluation datasets, we aim to intergrate datasets that closely resemble real-world vertical federated learning (VFL) applications. Specifically, we selected data with inherent characteristics that naturally originate from different clients. Our selection criteria prioritize data diversity across devices, domains, and modalities, ensuring they are both representative of practical VFL scenarios and suitable for our evaluation framework.
>
> > Minor issues. In Figure 2 and 3, there should be periods in the captions. In Table 19 and Table 20, missing spaces in table header of dataset name.
>
> Response: Thank you for your detailed suggestions. We greatly appreciate your thorough review and will carefully examine the errors.
>
> [1] Chong Fu, et al. Label inference attacks against vertical federated learning. In 31st USENIX security symposium (USENIX Security 22), 2022.

---

> > ### Comment · Reviewer_wZVx · 2025-08-05
> >
> > I appreciate the authors' response, which has addressed all of my concerns. As my initial rating was already the highest, I will maintain it.

---

### Official Review · Reviewer_ADRt · 2025-06-22

**Ethics Flags:** Data privacy, copyright, and consent
**Rating:** 4
**Confidence:** 4

**Summary:**

This paper proposes a benchmark, MARS-VFL, for Vertical Federated Learning (VFL) evaluation to address the public data lack issues. By integrating 12 different datasets across five real-world applications, this benchmark could be utilized to evaluate the performance of VFL algorithms on efficiency, robustness and security. This benchmark could contribute to the development of VFL. The paper also proposes a new method, RVFL, for addressing robustness challenges, which acts as a baseline method for future research.

**Additional Feedback:**

Given the importance of data heterogeneity in real-world applications, I suggest that future work leverage this benchmark to investigate and address this challenge thoroughly.

**Dataset Code Accessibility:**

Yes

**Dataset Code Comments:**

The paper provides a Git repository that includes tutorials for both the benchmark and the referenced algorithms.

**Ethical Comments:**

How does the dataset ensure that user privacy is preserved? Specifically, what concrete measures have been implemented to anonymize user identifiers or other sensitive information?

**Ethical Considerations:**

Yes, there are ethics concerns that require attention by the authors

**Final Justification:**

My questions and concerns have been addressed. I will maintain a positive score and look forward to further improvements in the paper.

**Limitations Weaknesses:**

- This work does not consider data heterogeneity and distribution imbalance, which are common and critical challenges in real-world scenarios. To enhance the applicability of VFL, it is essential to address these heterogeneity issues, and the benchmark should include corresponding evaluations.

- It appears that all algorithms are implemented using relatively simple network backbones, such as MLPs, CNNs, and LSTMs. These architectures are somewhat outdated and could be replaced with more advanced foundational models, such as DINO, CLIP, or other multi-modal large language models (MLLMs). Implementing the algorithms on larger models would significantly enhance the benchmark's impact and value.

- As a benchmark, the evaluation of baseline methods is somewhat limited. Although the benchmark includes 12 datasets, it would strengthen the work to incorporate a broader range of existing algorithms for a more comprehensive performance evaluation.

**Strengths Contributions:**

- The proposed benchmark includes 12 datasets spanning five practical applications, enabling the evaluation of VFL models in realistic scenarios. By offering diverse and representative datasets, this benchmark helps address the data scarcity that has hindered the development of vertical federated learning.

- This paper provides a baseline method to address the robustness problem in VFL.

- The provided code and tutorials are clear and well-organized, making the benchmark easy to follow and reproduce.

---

> ### Author Rebuttal · Authors · 2025-07-30
>
> Thank you for your valuable comments and positive feedback. Below, we provide point-by-point responses to your suggestions. We welcome further discussion on any of these topics:
>
> > This work does not consider data heterogeneity and distribution imbalance, which are common and critical challenges in real-world scenarios. To enhance the applicability of VFL, it is essential to address these heterogeneity issues, and the benchmark should include corresponding evaluations.
>
> Response: Thanks for raising the critical issue. Exactly, data distributions can indeed be heterogeneous across clients. For instance, class distributions may vary between clients, or data quality may differ due to variations in collection capabilities.
>
> Currently, evaluating vertical federated learning (VFL) with heterogeneous class distributions is challenging due to the limitations of existing datasets, unlike in traditional horizontal federated learning. However, our framework can assess data quality heterogeneity by applying different levels of perturbations to different clients, which is already supported in the current version. Additionally, our benchmark incorporates diverse data types across clients, which inherently introduces data heterogeneity.
>
> We will carefully expand and update our benchmark to better address various aspects of data heterogeneity. Thank you again for your valuable suggestions—they are greatly appreciated as we continue to improve this work.
>
> > It appears that all algorithms are implemented using relatively simple network backbones, such as MLPs, CNNs, and LSTMs. These architectures are somewhat outdated and could be replaced with more advanced foundational models, such as DINO, CLIP, or other multi-modal large language models (MLLMs). Implementing the algorithms on larger models would significantly enhance the benchmark's impact and value.
>
> Response: Thank you for your insightful comments. Following the experimental settings of existing works [1][2][3], we primarily integrate and evaluate with the most commonly used backbones for fair comparison. Expanding our framework to include large-scale models such as CLIP and MLLMs is indeed an important direction for future updates.
>
> Here, we outline a potential approach for such an update: In cross-modality retrieval scenarios where both text and images must remain private, we could distribute the CLIP model's text encoder and image encoder across two different clients. The final retrieval operation would then be performed by the active client, which maintains the matching labels.
>
> To validate this concept, we conducted a preliminary two-client experiment with the pretrained CLIP models:
>
> - The active client hosts the CLIP text encoder, text descriptions, and matching labels
> - The passive client maintains the CLIP image encoder and corresponding images
> - Intermediate embeddings are transmitted to the active client for loss calculation and gradient updates
>
> The text-to-image retrievel results on the Flickr30K dataset are presented below:
>
> | Model            | R@1   | R@5   | MAP   |
> |------------------|-------|-------|-------|
> | *CLIP (ViT-B/32)*  | 56.31 | 81.25 | 66.33 |
> | *CLIP (ViT-B/16)*  | 60.73 | 83.59 | 71.28 |
>
> This setting can be extended to support various models including MLLMs, and we will continue to implement these updates in future versions.
>
> > As a benchmark, the evaluation of baseline methods is somewhat limited. Although the benchmark includes 12 datasets, it would strengthen the work to incorporate a broader range of existing algorithms for a more comprehensive performance evaluation.
>
> Response: Thank you for raising this important issue. Given the limitations of existing works, our current version includes the most recent and representative baselines to provide an easily extensible benchmark for future expansion. We will continue to update these baselines to incorporate future advances in vertical federated learning (VFL).
>
> > How does the dataset ensure that user privacy is preserved? Specifically, what concrete measures have been implemented to anonymize user identifiers or other sensitive information?
>
> Response: Thank you for your constructive suggestions. For future algorithm deployments, existing privacy-preserving protocols can be easily integrated into the framework to ensure user privacy [4], as preliminarily discussed in Section 2.3 of our paper. We will expand related discussions more thoroughly following your suggestions.
>
> [1] Chong Fu, et al. Label inference attacks against vertical federated learning. In 31st USENIX security symposium (USENIX Security 22), 2022.
>
> [2] Tianyuan Zou, et al. Vflair: A research library and benchmark for vertical federated learning. In International Conference on Learning Representations, 2024.
>
> [3] Pedro Valdeira, et al. Vertical federated learning with missing features during training and inference. In International Conference on Learning Representations, 2025.
>
> [4] Stephen Hardy, et al. Private federated learning on vertically partitioned data via entity resolution and additively homomorphic encryption. arXiv, 2017.

---

> > ### Comment · Reviewer_ADRt · 2025-08-05
> >
> > Thank you for your response. It answers my questions and concerns, so I will be keeping my positive score.

---

### Official Review · Reviewer_3hSL · 2025-06-28

**Rating:** 4
**Confidence:** 4

**Summary:**

This work focuses a critical and timely gap in the evaluation of vertical federated learning. While previous studies typically rely on synthetic data or manually constructed data splits, this work puts forward to using data from practical scenes. It brings the evaluation closer to practical settings. The paper offers complete evaluations across three essential dimensions: efficiency, robustness, and security, with a novel baseline introduced. The extensive experiments and analysis are conducted, offering insights for evaluating related methods.

**Additional Feedback:**

N/A

**Dataset Code Accessibility:**

Yes

**Dataset Code Comments:**

Details for reproducing are contained in the paper. The code is well-documented, with datasets publicly accessible.

**Ethical Considerations:**

No, there are no or only very minor ethics concerns

**Final Justification:**

Thanks for the author's rebuttal, and all my concerns have been addressed. Thus, I will keep my score.

**Limitations Weaknesses:**

Some Questions:

- Scalability limitations with respect to client scales: The framework supports a maximum of five clients due to constraints in data structure. Many applications—such as those in finance, healthcare, or large-scale industry—often involve a much greater number of participants. It would be helpful to explore whether the framework could be extended to accommodate more clients. For example, could alternative data splitting approaches or more scalable collaboration protocols be introduced to support larger scales?

- Computational overhead of the proposed RVFL: It introduces additional time complexity, especially due to the cubic time complexity of the RVFL-Align. This could limit its feasibility in large-scale or latency-sensitive settings. The paper would benefit from more details of this complexity, such as through the underlying matching algorithm—and an analysis of the associated communication cost. Additionally, exploring potential strategies to reduce the computational burden, such as approximate matching or parallelization, would make it more practical for wider adoption.

- The simulation of corrupted features: To generate feature corruptions, A random Gaussian noise is added to the origin features. However, the corruptions should come from many kinds. For example, image data may be cropped or blurry. Are more corruptions accessible?

**Strengths Contributions:**

This framework serves as a valuable tool for researchers and practitioners, particularly in situations where practical data is difficult to access or collect. The paper introduces a novel robustness baseline, which have not been thoroughly explored in prior work. The framework is significant from both engineering and technical perspectives, helping to narrow the gap in existing works.

---

> ### Author Rebuttal · Authors · 2025-07-30
>
> Thank you for your valuable feedback and insightful comments. We greatly appreciate your thoughtful input, which helps us improve our work. We present our responses below:
>
> > Scalability limitations with respect to client scales: The framework supports a maximum of five clients due to constraints in data structure. Many applications—such as those in finance, healthcare, or large-scale industry—often involve a much greater number of participants. It would be helpful to explore whether the framework could be extended to accommodate more clients. For example, could alternative data splitting approaches or more scalable collaboration protocols be introduced to support larger scales?
>
> Response: Thank you for your thoughtful comments. Exactly, many real-world applications, particularly in domains like finance, healthcare, and large-scale industry, require support for a significantly higher number of participants.
>
> Currently, the framework’s five-client limit is indeed a constraint imposed by its centralized data structure and collaboration protocol design. However, we recognize the importance of scalability and are actively exploring ways to enhance the framework’s capabilities.
>
> Your suggestion about leveraging existing data-splitting approaches to extend the number of clients is excellent. Such methods could be easily integrated into our framework, building on existing work while expanding client support based on our data.
>
> > Computational overhead of the proposed RVFL: It introduces additional time complexity, especially due to the cubic time complexity of the RVFL-Align. This could limit its feasibility in large-scale or latency-sensitive settings. The paper would benefit from more details of this complexity, such as through the underlying matching algorithm—and an analysis of the associated communication cost. Additionally, exploring potential strategies to reduce the computational burden, such as approximate matching or parallelization, would make it more practical for wider adoption.
>
> Response: Thank you for raising this concern. We’d like to clarify that the primary cost introduced by RVFL-Align is computation time, with no additional communication overhead. We measured the time cost of RVFL-Align on the UCI-HAR dataset at different data scales by multiplying the original dataset size by 10× and 50×:
>
> **Table R1:**
>
> | Scale | Total Time | vs Base | Time Cost for HG | vs Base | Samples   |
> |-------|------------|---------|------------------|---------|-----------|
> | *Base*  | 336.52     | 1.00x   | 89.78            | 1.00x   | 10,299    |
> | *10x*   | 3,335.67   | 9.91x   | 827.16           | 9.21x   | 102,990   |
> | *50x*   | 21,826.92  | 64.86x  | 4,377.58         | 48.75x  | 514,950   |
>
> The time costs introduced by $O(N^3)$ complexity is the main bottleneck of RVFL-Align, especially with larger sample scales.
>
> For reduce the computational burden, we could utilize greedy matching algorithm, which realigns samples sequentially by selecting the maximum similarity at each step, thereby reducing the complexity to $O(N^2)$. However, this approach may converge to a local optimum. The results are presented below:
>
> **Table R2:**
>
> | Method        |                |   $r_a=0$    |       |              |   $r_a=0.8$    |       |                |   $r_a=1$    |       |
> |---------------|---------------------|-------|-------|---------------------|-------|-------|---------------------|-------|-------|
> |               | $r_b=0$   | $r_b=0.8$   | $r_b=1$   | $r_b=0$   | $r_b=0.8$   | $r_b=1$   | $r_b=0$   | $r_b=0.8$   | $r_b=1$   |
> | *Base*          | **94.67**   | 90.87   | 89.01 | 90.87   | 87.75   | 86.56 | 90.60    | 87.75   | 84.86 |
> | *Greedy-Matching*  | 94.02   | 90.57   | 90.87 | 90.22   | 89.38   | 88.19 | 89.79    | 88.26   | 86.39 |
> | *RVFL-Align*       | 94.47   | **90.91**   | **91.01** | **91.08**   | **91.08**   | **91.04** | **90.80**    | **91.08**   | **91.08** |
>
> > The simulation of corrupted features: To generate feature corruptions, A random Gaussian noise is added to the origin features. However, the corruptions should come from many kinds. For example, image data may be cropped or blurry. Are more corruptions accessible?
>
> Response: Thank you for your valuable feedback. For corrupted features, we implement a general approach using Gaussian noise, which can be applied to different data types. For other types of corruption, we can follow the common types appear in different data: For image data, the geometric deformations such as rotation, scale variations and cropping can be utilized; For text data, some of words could be spelled mistakenly, a random permutation of words could be introduced; For sensor data, sensor outages causing contiguous zeros, so we can randomly mask feature values for simulation.

---

> > ### Comment · Reviewer_3hSL · 2025-08-05
> >
> > Thanks for the author's rebuttal, and all my concerns have been addressed.

---

### Official Review · Reviewer_ktk4 · 2025-07-01

**Rating:** 6
**Confidence:** 5

**Summary:**

This paper presents MARS-VFL, a unified benchmark for Vertical Federated Learning (VFL), addressing the lack of standardized datasets and evaluation protocols that hinder fair comparison and reproducibility. It offers 12 curated datasets from five real-world domains where features are naturally partitioned (e.g., by device or modality), and a comprehensive evaluation framework covering efficiency, robustness, and security. A key contribution is the formalization of robustness issues—missing, corrupted, and misaligned features—and the proposal of RVFL to handle corrupted/misaligned data. Extensive experiments benchmark recent VFL methods. Overall, MARS-VFL sets a solid foundation for future VFL research.

**Additional Feedback:**

MARS-VFL squarely addresses a recognized bottleneck—the lack of realistic, reproducible, and multi-aspect benchmarks for VFL. The dataset curation, unified protocols, and initial baselines together constitute a substantial and lasting contribution that will shape future research directions. While some scope and polish issues remain, they are fixable and do not undermine the core value. I therefore recommend acceptance.

**Dataset Code Accessibility:**

Yes

**Dataset Code Comments:**

The authors have released their code, dataset, and protocols with documentation. The reproduction is straightforward.

**Ethical Considerations:**

No, there are no or only very minor ethics concerns

**Final Justification:**

All my concerns have been well addressed. This is excellent work that will strongly advance the Vertical Federated Learning community.

**Limitations Weaknesses:**

The paper is very strong, and the following points are primarily intended as constructive suggestions for further strengthening the work and guiding future iterations of the benchmark.

1). The authors correctly identify that the $O(N^3)$ time complexity of the Hungarian algorithm used in RVFL-Align is a potential limitation for large datasets. This discussion could be strengthened by providing more practical context. It would be beneficial to quantify the practical limits of this approach. For example, on the hardware used for the experiments, what is the approximate number of samples (N) at which the realignment becomes computationally prohibitive? Furthermore, the authors could suggest or briefly discuss potential approximation algorithms (e.g., greedy matching or mini-batch-based alignment) as a practical alternative for large-scale scenarios. This would provide a more complete picture for practitioners looking to adopt the method.

2). The paper honestly reports that RVFL can sometimes degrade performance on clean/well-aligned data. This is an important finding that warrants a slightly deeper investigation. Thus, a brief discussion on why this performance drop occurs would be insightful. For RVFL-Aug, is the information loss from augmentations inherent, or could it be mitigated with a different augmentation strategy or a more adaptive weighting of the consistency loss? For RVFL-Align, could a simple heuristic be introduced to prevent unnecessary shuffling? For instance, one could measure the initial alignment consistency and only trigger the realignment process if it falls below a certain threshold.

3). The robustness and security evaluations are comprehensive, forming a strong foundation. However, there is room for future expansion. The current benchmark uses Gaussian noise to simulate corrupted features. While standard, real-world corruptions can be more structured (e.g., sensor outages causing contiguous zeros, systematic biases). A brief discussion on how the framework could be extended to these more complex corruption types would be valuable. Furthermore, the evaluated defenses are general gradient-based techniques. The paper would be even more comprehensive if it included a discussion on defenses specifically designed for the VFL topology and its unique vulnerabilities, providing a clearer path for future research in VFL-specific security.

4). A significant number of the benchmark’s datasets, particularly in the multimedia and emotion analysis domains (e.g., NUS-WIDE, MM-IMDB, UR-FUNNY), utilize pre-extracted features rather than raw data. While this simplifies setup, it is a considerable weakness because it abstracts away a critical component of a real-world VFL system: local, end-to-end feature extraction. In practice, each client would run its own deep learning model (e.g., a CNN on raw images or a Transformer on raw text) to generate embeddings.  Future versions of the benchmark should prioritize incorporating tasks that operate on raw data, requiring each client to perform its own feature extraction. This would provide a more realistic assessment of the trade-offs between local computation, communication overhead, and overall task performance, which is central to the VFL paradigm.

5). The paper effectively models data heterogeneity by design, but it overlooks other critical forms of heterogeneity that are prevalent in real-world federated systems. The experiments are conducted on a powerful and homogeneous server environment, which does not account for system heterogeneity and model heterogeneity.

6). The robustness evaluation is well-structured, but it assesses each challenge—missing, corrupted, and misaligned features—in isolation. Real-world data is often messy in multiple ways simultaneously. For example, a data transmission error could corrupt some features of a sample while a database indexing error misaligns it with features from another client. The current benchmark does not evaluate how algorithms perform under these compound perturbations.

7). The proposed RVFL-Aug method for handling corrupted features relies on a set of generic data augmentations (random mask, scaling) applied across all data types. While this ensures broad applicability, it is also a potential weakness, as generic augmentations are often suboptimal compared to domain-specific ones. For instance, augmentations for time-series ECG data (e.g., time warping, magnitude warping) are very different from those for images (e.g., rotation, cropping) or text. Using a one-size-fits-all approach may not unlock the full potential of consistency learning for robustness.

**Strengths Contributions:**

1). The development of a unified benchmark directly addresses a pressing need within the VFL community. The lack of standardized evaluation environments has been a major obstacle to measuring progress. By providing a well-structured benchmark with publicly available code  and data, this work will undoubtedly foster more rigorous, comparable, and reproducible research.

2). The core design principle of MARS-VFL—using datasets that are naturally partitioned rather than artificially split—is a major advancement. This approach, using data from different sensors , modalities , or domains  as distinct clients, ensures that the evaluation scenarios closely mirror the practical challenges and data distributions of real-world VFL deployments.

3). The benchmark's scope is commendably broad. Instead of focusing on a single metric, it establishes a multi-faceted evaluation protocol covering three foundational pillars (efficiency, robustness, and security) of any practical machine learning system.

4).  This work is more than just a benchmark; it actively advances the field. The authors are the first to unify the evaluation of robustness challenges in VFL. Furthermore, they propose a novel and practical baseline, RVFL (with its RVFL-Aug and RVFL-Align variants), to address the under-explored problems of corrupted and misaligned features, providing a valuable starting point for future research in this critical area.

5). The empirical evaluation is extensive and rigorous. The authors benchmark multiple recent methods across the 12 datasets and provide detailed results, including performance under a wide range of conditions (e.g., varying perturbation rates). The inclusion of detailed appendices with model architectures, hyperparameters, and evaluation settings further enhances the paper's value and reproducibility. The authors are also transparent about the limitations of their own proposed methods.

6). The transparent limitations & future plans are explicitly discussed, showing awareness of benchmark extensibility and fairness gaps.

---

> ### Author Rebuttal · Authors · 2025-07-30
>
> We sincerely appreciate your constructive comments and positive feedback, which highlight several critical aspects for future work. We provide point-by-point responses below:
>
> > The authors correctly identify that the $O(N^3)$ time complexity of the Hungarian algorithm used in RVFL-Align is a potential limitation for large datasets. This discussion could be strengthened by providing more practical context. It would be beneficial to quantify the practical limits of this approach. For example, on the hardware used for the experiments, what is the approximate number of samples (N) at which the realignment becomes computationally prohibitive? Furthermore, the authors could suggest or briefly discuss potential approximation algorithms (e.g., greedy matching or mini-batch-based alignment) as a practical alternative for large-scale scenarios. This would provide a more complete picture for practitioners looking to adopt the method.
>
> Response:
> The main costs associated with the complexity of RVFL-Align are computing time and computational resource usage, while memory consumption varies across different datasets. Both time and resource costs are difficult to standardize, making it challenging to estimate the maximum value of $N$. In this case, we report the time cost on our hardware using the UCI-HAR dataset under the baseline settings mentioned in the paper. We conduct experiments with data repetitions of 10× and 50× the original dataset size:
>
> **Table R1:**
>
> | Scale | Total Time | vs Base | Time Cost for HG | vs Base | Samples   |
> |-------|------------|---------|------------------|---------|-----------|
> | *Base*  | 336.52     | 1.00x   | 89.78            | 1.00x   | 10,299    |
> | *10x*   | 3,335.67   | 9.91x   | 827.16           | 9.21x   | 102,990   |
> | *50x*   | 21,826.92  | 64.86x  | 4,377.58         | 48.75x  | 514,950   |
>
> As shown in the Table R2, the main limitation of RVFL-Align is its high computational complexity-$O(N^3)$, which could be optimized in future work to significantly reduce training time, thereby improving the algorithm's efficiency. This issue becomes more pronounced as $N$ increases.
>
> To address this, we implement a greedy-matching algorithm that realigns samples sequentially based on maximum similarity. We present the results on the UCI-HAR dataset below:
>
> **Table R2:**
>
> | Method        |                |   $r_a=0$    |       |              |   $r_a=0.8$    |       |                |   $r_a=1$    |       |
> |---------------|---------------------|-------|-------|---------------------|-------|-------|---------------------|-------|-------|
> |               | $r_b=0$   | $r_b=0.8$   | $r_b=1$   | $r_b=0$   | $r_b=0.8$   | $r_b=1$   | $r_b=0$   | $r_b=0.8$   | $r_b=1$   |
> | *Base*          | **94.67**   | 90.87   | 89.01 | 90.87   | 87.75   | 86.56 | 90.60    | 87.75   | 84.86 |
> | *Greedy-Matching*  | 94.02   | 90.57   | 90.87 | 90.22   | 89.38   | 88.19 | 89.79    | 88.26   | 86.39 |
> | *RVFL-Align*       | 94.47   | **90.91**   | **91.01** | **91.08**   | **91.08**   | **91.04** | **90.80**    | **91.08**   | **91.08** |
>
> As illustrated in Table R2, although it reducing the complexity to O(N²), greedy-matching may converge to a local optimum and a sub-optimal performance.
>
> > The paper honestly reports that RVFL can sometimes degrade performance on clean/well-aligned data. This is an important finding that warrants a slightly deeper investigation. Thus, a brief discussion on why this performance drop occurs would be insightful. For RVFL-Aug, is the information loss from augmentations inherent, or could it be mitigated with a different augmentation strategy or a more adaptive weighting of the consistency loss? For RVFL-Align, could a simple heuristic be introduced to prevent unnecessary shuffling? For instance, one could measure the initial alignment consistency and only trigger the realignment process if it falls below a certain threshold.
>
> Response: Thank you for your valuable suggestions.
>
> Regarding RVFL-Aug, we believe the performance degradation on clean data primarily stems from information loss during augmentation—a common phenomenon in noise-tolerant federated learning methods. This presents an interesting research direction, where new weighting strategies could be explored to mitigate such degradation.
>
> For RVFL-Align, the model may inadvertently shuffle well-aligned samples, leading to performance drops. A heuristic approach to prevent unnecessary shuffling of well-aligned data could be a promising avenue for future work, warranting deeper exploration and ablation studies. In this work, we establish a simple yet extensible robustness baseline, which can serve as a foundation for further research.
>
> > The robustness and security evaluations are comprehensive, forming a strong foundation. However, there is room for future expansion. The current benchmark uses Gaussian noise to simulate corrupted features. While standard, real-world corruptions can be more structured (e.g., sensor outages causing contiguous zeros, systematic biases). A brief discussion on how the framework could be extended to these more complex corruption types would be valuable. Furthermore, the evaluated defenses are general gradient-based techniques. The paper would be even more comprehensive if it included a discussion on defenses specifically designed for the VFL topology and its unique vulnerabilities, providing a clearer path for future research in VFL-specific security.
>
> Response: Thank you for highlighting this critical issue. For corrupted features, we implement a general version with Gaussian noise, which can be adapted to different data types. For other corruptions, we can follow  common patterns observed in various data types: Image data: Geometric deformations (e.g., rotation, scaling, cropping); Text data: Misspelled words or random word permutations; Sensor data: Simulated sensor outages (e.g., contiguous zeros or randomly masked feature values).
>
> For evauations with VFL defenses, given the limited recent research on VFL defenses, we adopted settings from open-source works like PMC and AMC [1]. Our framework is highly extensible, making it straightforward to incorporate future VFL-specific defense strategies.
>
> [1] Chong Fu, et al. Label inference attacks against vertical federated learning. In 31st USENIX security symposium (USENIX Security 22), 2022.
>
> > A significant number of the benchmark's datasets, particularly in the multimedia and emotion analysis domains (e.g., NUS-WIDE, MM-IMDB, UR-FUNNY), utilize pre-extracted features rather than raw data. While this simplifies setup, it is a considerable weakness because it abstracts away a critical component of a real-world VFL system: local, end-to-end feature extraction. In practice, each client would run its own deep learning model (e.g., a CNN on raw images or a Transformer on raw text) to generate embeddings. Future versions of the benchmark should prioritize incorporating tasks that operate on raw data, requiring each client to perform its own feature extraction.
>
> Response: We agree that existing datasets (e.g., NUS-WIDE, MM-IMDB) relying on pre-extracted features fail to address a critical challenge in real-world VFL systems: local end-to-end feature extraction. In this work, we establish a simplified setting for realistic VFL evaluation while using pre-extracted features to streamline experiments. In future work, we plan to continuously update the benchmark by incorporating client-side feature extraction settings.
>
> > The paper effectively models data heterogeneity by design, but it overlooks other critical forms of heterogeneity that are prevalent in real-world federated systems. The experiments are conducted on a powerful and homogeneous server environment, which does not account for system heterogeneity and model heterogeneity.
>
> Response: Thank you for your comments. Regarding model heterogeneity, our benchmark already accounts for this scenario by incorporating different model architectures for distinct data types across clients (e.g., using specialized models for different sensor data). For system heterogeneity, we currently follow existing homogeneous settings without further exploration. Investigating diverse system environments across clients would indeed be a valuable direction for future research.
>
> > The robustness evaluation is well-structured, but it assesses each challenge—missing, corrupted, and misaligned features—in isolation. Real-world data is often messy in multiple ways simultaneously. For example, a data transmission error could corrupt some features of a sample while a database indexing error misaligns it with features from another client. The current benchmark does not evaluate how algorithms perform under these compound perturbations.
>
> Response: Thank you for highlighting this issue. The exploration of robustness in VFL remains at an early stage. Indeed, real-world data often contains multiple simultaneous perturbations, making it essential to develop unified methods for handling such mixed scenarios. While our benchmark is extensible for evaluating multiple perturbations, the proposed RVFL was not specifically designed for these complex cases. Developing effective methods and evaluation metrics for such situations will be an important direction for future research.
>
> > The proposed RVFL-Aug method for handling corrupted features relies on a set of generic data augmentations (random mask, scaling) applied across all data types. While this ensures broad applicability, it is also a potential weakness, as generic augmentations are often suboptimal compared to domain-specific ones.
>
> Response: Thanks for the constructive advice. The proposed RVFL utilizes a unified augmentation method for different types of data, which shows generalization and effectiveness for different data types. Using specific augmentations for different types of data could be further enhanced the performance, and can be further explored.

---

> > ### Comment · Reviewer_ktk4 · 2025-08-08
> >
> > Thank you for your thoughtful reply. After carefully reviewing the rebuttal, I find that all my concerns have been fully addressed.

---

### Note · Authors · 2025-08-13

**Summary:**

We sincerely thank all reviewers for their constructive comments, insightful suggestions, and recognition of our contributions. We appreciate the positive feedback, noting that our work: (1) offers an extensible framework relevant to VFL applications with realistic evaluations. (2) first unifies robustness challenges in VFL with a novel baseline; (3) transparently discusses limitations and future works. We provide the key responses and acknowledge the ethical impacts, as follows:

**Key Responses:**
- *RVFL-Align Overhead:* The main overhead is computational time from the $O(N^3)$ Hungarian algorithm. We provide timing analyses and a greedy-matching alternative ($O(N^2)$) to trade optimality for efficiency. A comparison with alternative alignments using greedy matching is provided to demonstrate the significance of the proposed RVFL-Align.
- *Benchmark Updates & Heterogeneity:* We will incorporate end-to-end local feature extraction and expanded dataset coverage, ensuring modality, domain, and partition diversity. The framework already supports certain heterogeneity forms and can extend to system-level heterogeneity and compound perturbations.
- *Future Extensions:* Plans include integrating larger backbones (e.g., CLIP, MLLMs), expanding baseline coverage, and enhancing perturbation realism.

**Ethics Impact:**
We adhere to privacy-preserving protocols for human-derived datasets, acknowledge potential misuse risks, and note the low environmental footprint of our experiments.

We believe our work establishes an extensible foundation for realistic, robust, and secure VFL research and remain committed to addressing the identified limitations in future iterations.

---

### Decision · Program_Chairs · 2025-09-18

**Decision:**

Accept (spotlight)

**Comment:**

The reviewers unanimously agree that the paper makes significant contributions to the benchmarking and evaluation effort of the field of vertical federated learning.

===== FINAL UPDATE FROM DB Track PCs ====

The final decision for this paper has been taken by the program chairs after consultation with the SACs. All Senior Area Chairs have ranked papers according to the feedback from the AC during the review process. We decided to leave the original meta-review to reflect the opinion of the AC in light of the initial discussions with reviewers and SAC.